# Towards a Scalable Reference-Free Evaluation of Generative Models

**Azim Ospanov**[*]
aospanov9@cse.cuhk.edu.hk

**Jingwei Zhang**[*]
jwzhang22@cse.cuhk.edu.hk

**Mohammad Jalali**[*]
mjalali24@cse.cuhk.edu.hk

**Xuenan Cao** [†]
xuenancao@cuhk.edu.hk

**Andrej Bogdanov** [‡]
abogdano@uottawa.ca

**Farzan Farnia**[*]
farnia@cse.cuhk.edu.hk

## Abstract

While standard evaluation scores for generative models are mostly reference-based, a reference-dependent assessment of generative models could be generally difficult due to the unavailability of applicable reference datasets. Recently, the reference-free entropy scores, VENDI [1] and RKE [2], have been proposed to evaluate the diversity of generated data. However, estimating these scores from data leads to significant computational costs for large-scale generative models. In this work, we leverage the random Fourier features framework to reduce the computational price and propose the *Fourier-based Kernel Entropy Approximation (FKEA)* method. We utilize FKEA's approximated eigenspectrum of the kernel matrix to efficiently estimate the mentioned entropy scores. Furthermore, we show the application of FKEA's proxy eigenvectors to reveal the method's identified modes in evaluating the diversity of produced samples. We provide a stochastic implementation of the FKEA assessment algorithm with a complexity $O(n)$ linearly growing with sample size $n$. We extensively evaluate FKEA's numerical performance in application to standard image, text, and video datasets. Our empirical results indicate the method's scalability and interpretability applied to large-scale generative models. The codebase is available at `https://github.com/aziksh-ospanov/FKEA`.

## 1 Introduction

A quantitative comparison of generative models requires evaluation metrics to measure the quality and diversity of the models' produced data. Since the introduction of variational autoencoders (VAEs) [3], generative adversarial networks (GANs) [4], and diffusion models [5] that led to impressive empirical results in the last decade, several evaluation scores have been proposed to assess generative models learned by different training methods and architectures. Due to the key role of evaluation criteria in comparing generative models, they have been extensively studied in the literature.

While various statistical methods have been applied to measure the fidelity and variety of a generative model's produced data, the standard scores commonly perform a reference-based evaluation of generative models, i.e., they quantify the characteristics of generated samples in comparison to a reference distribution. The reference distribution is usually chosen to be either the distribution of

---

[*]Department of Computer Science & Engineering, The Chinese University of Hong Kong, Hong Kong

[†]Department of Cultural and Religious Studies, The Chinese University of Hong Kong, Hong Kong

[‡]School of Electrical Engineering and Computer Science, University of Ottawa, Canada

38th Conference on Neural Information Processing Systems (NeurIPS 2024).

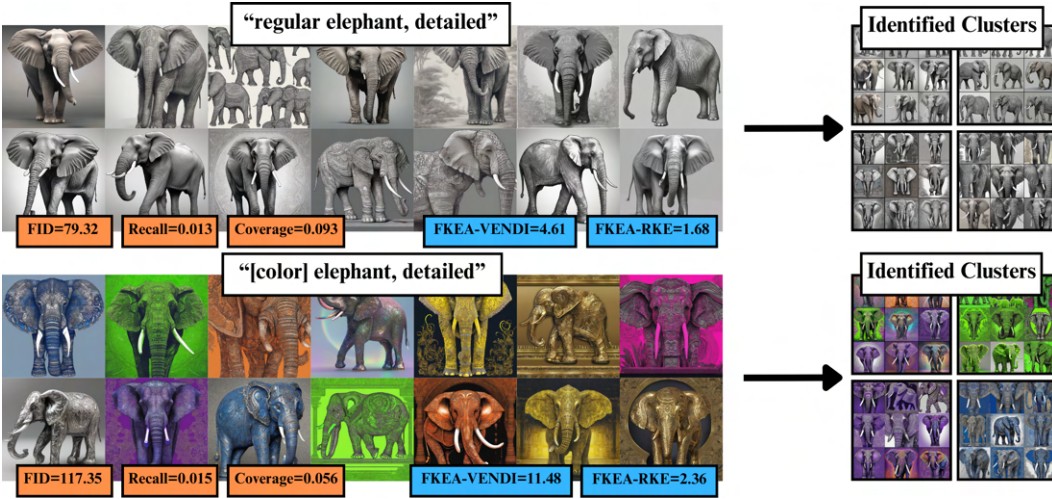

Figure 1: Reference-based vs. reference-free scores on two datasets of Stable Diffusion XL generated elephant images. FID, Recall, and Coverage scores (colored orange) are reference-based, whereas VENDI and RKE scores (colored blue) are reference-free. Inception.V3 is used as the backbone embedding. Reference-based metrics use 'Indian elephant' samples in ImageNet as reference data.

samples in the test data partition or a comprehensive dataset containing a significant fraction of real-world sample types, e.g. ImageNet [6] for evaluating image-based generative models.

To provide well-known examples of reference-dependent metrics, note that the distance scores, Fréchet Inception Distance (FID) [7] and Kernel Inception Distance (KID) [8], are explicitly reference-based, measuring the distance between the generative and reference distributions. Similarly, the standard quality/diversity score pairs, Precision/Recall [9, 10] and Density/Coverage [11], perform the evaluation in comparison to a reference dataset. Even the seemingly reference-free Inception Score (IS) [12] can be viewed as implicitly reference-based, since it quantifies the variety and fidelity of data based on the labels and confidence scores assigned by an ImageNet pre-trained neural net, where ImageNet implicitly plays the role of the reference dataset. The reference-based nature of these evaluation scores is desired in many instances including standard image-based generative models, where either a sufficiently large test set or a comprehensive reference dataset such as ImageNet is available for the reference-based evaluation.

On the other hand, a reference-based assessment of generative models may not always be feasible, because the selection of a reference distribution may be challenging in a general learning scenario. For example, in prompt-based generative models where the data are created in response to a user's input text prompts, the generated data could follow an a priori unknown distribution depending on the specific distribution of the user's input prompts. Figure 1 shows one such example where we compare reference-based diversity scores of regular and colored elephant image samples generated by Stable Diffusion XL [13]. While the diversity of the colored images looks significantly higher to the human eye, the evaluated reference-based FID, Recall, and Coverage metrics do not suggest a higher diversity. As this example suggests, a proper reference-based evaluation of every user's generated data would require a distinct reference dataset, which may not be available to the user during the assessment time. Moreover, finding a comprehensive text or video dataset to choose as the reference set would be more difficult compared to image data, because the higher length of text and video samples could significantly contribute to their variety, requiring an inefficiently large reference set to cover all text or video sample types.

The discussed challenging scenarios of conducting a reference-based evaluation highlight the need for reference-free assessment methods that remain functional in the absence of a reference dataset. Recently, entropy-based diversity evaluation scores, the VENDI metric family [1, 14] and RKE score [2], have been proposed to address the need for reference-free assessment metrics. These scores calculate the entropy of the eigenvalues of a kernel similarity matrix for the generated data. Based on the theoretical results in [2], the evaluation process of these scores can be interpreted as an unsupervised identification of the generative model's produced sample clusters, followed by the entropy calculation for the frequencies of the detected clusters. In Figure 1, we observe that the

reference-free VENDI and RKE scores grow when the generated samples are colored, which is due to the increase in the quantity of identified clusters in the colored case.

While the VENDI and RKE entropy scores provide reference-free assessments of generative models, estimating these scores from generated data could incur significant computational costs. In this work, we show that computing the precise RKE and VENDI scores would require at least $\Omega(n^2)$ and $\Omega(n^{2.373})^4$ computations for a sample size $n$, respectively. While the randomized projection methods in [15, 1] can reduce the computational costs to $O(n^2)$ for a general VENDI$_\alpha$ score, the quadratic growth would be a barrier to the method's application to large $n$ values. Although the computational expenses could be reduced by limiting the sample size $n$, an insufficient sample size would lead to significant error in estimating the entropy scores. As an example on the ImageNet dataset, Figure 7 in the Appendix shows the adverse effects of limiting the sample size on the quality of clusters used in the calculation of the VENDI scores.

To overcome the challenges of computing the scores, we leverage the random Fourier features (RFFs) framework [16] and develop a scalable entropy-based evaluation method that can be efficiently applied to large sample sizes. Our proposed method, *Fourier-based Kernel Entropy Approximation (FKEA)*, is designed to approximate the kernel covariance matrix using the RFFs drawn from the Fourier transform-inverse of a target shift-invariant kernel. We prove that using a Fourier feature size $r = \mathcal{O}\left(\frac{\log n}{\epsilon^2}\right)$, FKEA computes the eigenspace of the kernel matrix within an $\epsilon$-bounded error. Furthermore, we demonstrate the application of the eigenvectors of the FKEA's proxy kernel matrix for identifying the sample clusters used in the reference-free evaluation of entropic diversity.

Finally, we present numerical results of the entropy-based evaluation of standard generative models using the baseline eigendecomposition and our proposed FKEA methods. In our experiments, the baseline spectral decomposition algorithm could not efficiently scale to sample sizes above a few ten thousand. On the other hand, our stochastic implementation of the FKEA method could scalably apply to large sample sizes. Utilizing the standard embeddings of image, text, and video data, we tested the FKEA assessment while computing the sample clusters and their frequencies in application to large-scale datasets and generative models. Here is a summary of our work's main contributions:

- Characterizing the computational complexity of the kernel entropy scores of generative models,
- Developing the Fourier-based FKEA method to approximate the kernel covariance eigenspace and entropy of generated data,
- Proving guarantees on FKEA's required size of random Fourier features indicating a complexity logarithmically growing with the dataset size,
- Providing numerical results on FKEA's reference-free assessment of large-scale image,text, video-based datasets and generative models.

## 2 Related Work

**Evaluation of deep generative models.** The assessment of generative models has been widely studied in the literature. The existing scores either quantify a distance between the distributions of real and generated data, as in FID [7] and KID [8] scores, or attempt to measure the quality and diversity of the trained generative models, including the Inception Score [12], quality/diversity metric pairs Precision/Recall [9, 10] and Density/Coverage [11]. The mentioned scores are reference-based, while in this work we focus on reference-free metrics. Also, we note that the evaluation of memorization and novelty has received great attention, and several scores including the authenticity score [17], the feature likelihood divergence [18], and the rarity score [19] have been proposed to quantify the generalizability and novelty of generated samples. Note that the evaluation of novelty and generalization is, by nature, reference-based. On the other hand, our study focuses on the diversity of data which can be evaluated in a reference-free way as discussed in [1, 2].

**Role of embedding in quantitative evaluation**. Following the discussion in [20], we utilize DinoV2 [21] image embeddings in most of our image experiments, as [20]'s results indicate DinoV2 can yield scores more aligned with the human notion of diversity. As noted in [22], it is possible to utilize other non-ImageNet feature spaces such as CLIP [23] and SwAV [24] as opposed to InceptionV3 [25] to

---

[4]This computation complexity is the minimum known achievable cost for multiplying $n \times n$ matrices which we prove to lower-bound the complexity of computing matrix-based entropy scores.

further improve metrics such as FID. In this work, we mainly focus on DinoV2 feature space, while we note that other feature spaces are also compatible with entropy-based diversity evaluation.

**Diversity assessment for text-based models.** To quantify the diversity of text data, the n-gram-based methods are commonly used in the literature. A well-known metric is the BLEU score [26], which is based on the geometric average of n-gram precision scores times the Brevity Penalty. To adapt BLEU score to measure text diversity, [27] proposes the Self-BLEU score, calculating the average BLEU score of various generated samples. To further isolate and measure diversity, N-Gram Diversity scores [28, 29, 30] were proposed and defined by a ratio between the number of unique n-grams and overall number of n-grams in the text. Other prominent metrics include Homogenization (ROUGE-L) [31], FBD [32] and Compression Ratios [33].

**Kernel PCA, Spectral Cluttering, and Random Fourier Features**. Kernel PCA [34] is a well-studied method of dimensionality reduction that utilizes the eigendecomposition of the kernel matrix, similar to the kernel-based diversity evaluation methods in [1, 2]. The related papers [35, 36] study the connections between kernel PCA and spectral clustering. Also, the analysis of random Fourier features [16] for performing scalable kernel PCA has been studied in [37, 38, 39, 40, 41]. We note that while the mentioned works characterize the complexity of estimating the eigenvectors, our analysis focuses on the complexity of computing the kernel matrix's eigenvalues via Fourier features, as we primarily seek to quantify the diversity of generated data using the kernel matrix's eigenvalues.

## 3   Preliminaries

Consider a generative model $\mathcal{G}$ generating random samples $\mathbf{x}_1, \ldots, \mathbf{x}_n \in \mathbb{R}^d$ following the model's probability distribution $P_{\mathcal{G}}$. In our analysis, we assume the $n$ generated samples are independently drawn from $P_{\mathcal{G}}$. Note that in VAEs [3] and GANs [4], the generative model $\mathcal{G}$ is a deterministic function $G : \mathbb{R}^r \to \mathbb{R}^d$ mapping an $r$-dimensional latent random vector $\mathbf{Z} \sim P_Z$ from a known distribution $P_Z$ to $G(\mathbf{Z})$ distributed according to $P_{\mathcal{G}}$. On the other hand, in diffusion models, $\mathcal{G}$ represents an iterative random process that generates a sample from $P_{\mathcal{G}}$. The goal of a sample-based diversity evaluation of generative model $\mathcal{G}$ is to quantify the variety of its generated data $\mathbf{x}_1, \ldots, \mathbf{x}_n$.

### 3.1   Kernel Function, Kernel Covariance Matrix, and Matrix-based Rényi Entropy

Following standard definitions, $k : \mathbb{R}^d \times \mathbb{R}^d \to \mathbb{R}$ is called a kernel function if for every integer $n \in \mathbb{N}$ and set of inputs $\mathbf{x}_1, \ldots, \mathbf{x}_n \in \mathbb{R}^d$, the kernel similarity matrix $K = \big[ k(\mathbf{x}_i, \mathbf{x}_j) \big]_{n \times n}$ is positive semi-definite. We call a kernel function $k$ normalized if for every input $\mathbf{x}$ we have $k(\mathbf{x}, \mathbf{x}) = 1$. A well-known example of a normalized kernel function is the Gaussian kernel $k_{\text{Gaussian}(\sigma^2)}$ with bandwidth parameter $\sigma^2$ defined as

$$k_{\text{Gaussian}(\sigma^2)}\big(\mathbf{x}, \mathbf{x}'\big) := \exp\Big(-\frac{\big\| \mathbf{x} - \mathbf{x}' \big\|_2^2}{2\sigma^2}\Big)$$

For every kernel function $k$, there exists a feature map $\phi : \mathbb{R}^d \to \mathbb{R}^m$ such that $k(\mathbf{x}, \mathbf{x}') = \langle \phi(\mathbf{x}), \phi(\mathbf{x}') \rangle$ is the inner product of the $m$-dimensional feature maps $\phi(\mathbf{x})$ and $\phi(\mathbf{x}')$. Given a kernel $k$ with feature map $\phi$, we define the kernel covariance matrix $C_X \in \mathbb{R}^{m \times m}$ of a distribution $P_X$ as

$$C_X := \mathbb{E}_{\mathbf{X} \sim P_X}\Big[\phi(\mathbf{X})\phi(\mathbf{X})^\top\Big] = \int p_X(\mathbf{x})\phi(\mathbf{x})\phi(\mathbf{x})^\top \mathrm{d}\mathbf{x}$$

The above matrix $C_X$ is positive semi-definite with non-negative values. Furthermore, assuming a normalized kernel $k$, it can be seen that the eigenvalues of $C_X$ will add up to 1 (i.e., it has unit trace $\text{Tr}(C_X) = 1$), providing a probability model. Therefore, one can consider the entropy of $C_X$'s eigenvalues as a quantification of the diversity of distribution $P_X$ based on the kernel similarity score $k$. Here, we review the general family of Rényi entropy used to define VENDI and RKE scores.

**Definition 1.** *For a positive semi-definite matrix $C_X \in \mathbb{R}^{m \times m}$ with eigenvalues $\lambda_1, \ldots, \lambda_m$, the order-$\alpha$ Rényi entropy $H_\alpha(C_X)$ for $\alpha > 0$ is defined as*

$$H_\alpha(C_X) := \frac{1}{1 - \alpha} \log\Big(\sum_{i=1}^{m} \lambda_i^\alpha\Big)$$

To estimate the entropy scores from finite empirical samples $\mathbf{x}_1, \ldots, \mathbf{x}_n$, we consider the empirical kernel covariance matrix $\widehat{C}_X$ defined as $\widehat{C}_X := \frac{1}{n} \sum_{i=1}^{n} \phi(\mathbf{x}_i) \phi(\mathbf{x}_i)^\top$. This matrix provides an empirical estimation of the population kernel covariance matrix $C_X$.

It can be seen that the $m \times m$ empirical matrix $\widehat{C}_X$ and normalized kernel matrix $\frac{1}{n} K = \frac{1}{n} [k(\mathbf{x}_i, \mathbf{x_j})]_{n \times n}$ share the same non-zero eigenvalues. Therefore, to compute the matrix-based entropy of the empirical covariance matrix $\widehat{C}_X$, one can equivalently compute the entropy of the eigenvalues of the kernel similarity matrix $K$. This approach results in the definition of the VENDI and RKE diversity scores: [1] defines the family of VENDI scores as

$$\mathrm{VENDI}_\alpha(\mathbf{x}_1, \ldots, \mathbf{x}_n) := \exp\Big(H_\alpha\big(\frac{1}{n} K\big)\Big) = \Big(\sum_{i=1}^{n} \lambda_i^\alpha\Big)^{\frac{1}{1-\alpha}},$$

where $\lambda_1, \ldots, \lambda_n$ denote the eigenvalues of the kernel matrix $\frac{1}{n} K$. Also, [2] proposes the RKE score, which is the special order-2 Renyi entropy, $\mathrm{RKE}(\mathbf{x}_1, \ldots, \mathbf{x}_n) = \exp(H_2(\frac{1}{n} K))$. To compute RKE without computing the eigenvalues, [2] points out the RKE score reduces to the Frobenius norm $\|\cdot\|_F$ of the kernel matrix as follows:

$$\mathrm{RKE}(\mathbf{x}_1, \ldots, \mathbf{x}_n) = \Big\|\frac{1}{n} K\Big\|_F^{-2} = \Big(\frac{1}{n^2} \sum_{i=1}^{n} \sum_{j=1}^{n} k(\mathbf{x}_i, \mathbf{x}_j)^2\Big)^{-1}$$

## 3.2   Shift-Invariant Kernels and Random Fourier Features

A kernel function $k$ is called shift-invariant, if there exists a function $\kappa : \mathbb{R}^d \to \mathbb{R}$ such that $k(\mathbf{x}, \mathbf{x}') = \kappa(\mathbf{x} - \mathbf{x}')$ for every $\mathbf{x}, \mathbf{x}' \in \mathbb{R}^d$. Bochner's theorem proves that a function $\kappa : \mathbb{R}^d \to \mathbb{R}$ will lead to a shift-invariant kernel similarity score $\kappa(\mathbf{x} - \mathbf{x}')$ between $\mathbf{x}, \mathbf{x}'$ *if and only if* its Fourier transform $\widehat{\kappa} : \mathbb{R}^d \to \mathbb{R}$ is non-negative everywhere (i.e, $\widehat{\kappa}(\boldsymbol{\omega}) \geq 0$ for every $\boldsymbol{\omega}$). Note that the Fourier transform $\widehat{\kappa}$ is defined as

$$\widehat{\kappa}(\boldsymbol{\omega}) := \frac{1}{(2\pi)^d} \int \kappa(\mathbf{x}) \exp(-i\boldsymbol{\omega}^\top \mathbf{x}) \mathrm{d}\mathbf{x}$$

Specifically, Bochner's theorem shows the Fourier transform $\widehat{\kappa}$ of a normalized shift-invariant kernel $k(\mathbf{x}, \mathbf{x}') = \kappa(\mathbf{x} - \mathbf{x}')$, where $\kappa(0) = 1$, will be a probability density function (PDF). The framework of random Fourier features (RFFs) [16] utilizes independent samples drawn from PDF $\widehat{\kappa}$ to approximate the kernel function. Here, given independent samples $\boldsymbol{\omega}_1, \ldots, \boldsymbol{\omega}_r \sim \widehat{\kappa}$, we form the following proxy feature map $\widetilde{\phi}_r : \mathbb{R}^d \to \mathbb{R}^{2r}$

$$\widetilde{\phi}_r(\mathbf{x}) = \frac{1}{\sqrt{r}} \Big[\cos(\boldsymbol{\omega}_1^\top \mathbf{x}), \sin(\boldsymbol{\omega}_1^\top \mathbf{x}), \ldots, \cos(\boldsymbol{\omega}_r^\top \mathbf{x}), \sin(\boldsymbol{\omega}_r^\top \mathbf{x})\Big]. \tag{1}$$

As demonstrated in [16, 42], the $2r$-dimensional proxy map $\widetilde{\phi}_r$ can approximate the kernel function as $k(\mathbf{x}, \mathbf{x}') = \mathbb{E}_{\boldsymbol{\omega} \sim \widehat{\kappa}} \Big[\cos(\boldsymbol{\omega}^\top \mathbf{x}) \cos(\boldsymbol{\omega}^\top \mathbf{x}') + \sin(\boldsymbol{\omega}^\top \mathbf{x}) \sin(\boldsymbol{\omega}^\top \mathbf{x}')\Big] \approx \widetilde{\phi}_r(\mathbf{x})^\top \widetilde{\phi}_r(\mathbf{x}')$.

## 4   Computational Complexity of VENDI & RKE Scores

As discussed, computing RKE and general $\mathrm{VENDI}_\alpha$ scores requires computing the order-$\alpha$ entropy of kernel matrix $\frac{1}{n} K$. Using the standard definition of $\alpha$-norm $\|\mathbf{v}\|_\alpha = \big(\sum_{i=1}^{n} |v_i|^\alpha\big)^{1/\alpha}$, we observe that the computation of $\mathrm{VENDI}_\alpha$ score is equivalent to computing the $\alpha$-norm $\|\boldsymbol{\lambda}\|_\alpha$ of the $n$-dimensional eigenvalue vector $\boldsymbol{\lambda} = [\lambda_1, \ldots, \lambda_n]$ where $\lambda_1 \geq \cdots \geq \lambda_n$ are the sorted eigenvalues of the normalized kernel matrix $\frac{1}{n} K$.

In the following theorem, we prove that except order $\alpha = 2$, which is the RKE score, computing any other $\mathrm{VENDI}_\alpha$ score is at least as expensive as computing the product of two $n \times n$ matrices. Therefore, the theorem suggests that the computational complexity of every member of the VENDI family is lower-bounded by $\Omega(n^{2.372})$ which is the least known cost of multiplying $n \times n$ matrices.

In the theorem, we suppose $\mathcal{B}$ is any fixed set of "basis" functions. A circuit $\mathcal{C}$ is a directed acyclic graph each of whose internal nodes is labeled by a *gate* coming from a set $\mathcal{B}$. A subset of gates are

designated as outputs of $\mathcal{C}$. A circuit with $n$ source nodes and $m$ outputs computes a function from $\mathbb{R}^n$ to $\mathbb{R}^m$ by evaluating the gate at each internal gate in topological order. The size of a circuit is the number of gates. Also, $\nabla \mathcal{B}$ is the basis consisting of the gradients of all functions in $\mathcal{B}$. We will provide the proof of the theorems in the Appendix.

**Theorem 1.** *If* $\mathrm{VENDI}_\alpha(K)$ *for* $\alpha \neq 2$ *is computable by a circuit* $\mathcal{C}$ *of size* $s(n)$ *over basis* $\mathcal{B}$, *then* $n \times n$ *matrices can be multiplied by a circuit* $\mathcal{C}$ *of size* $O(s(n))$ *over basis* $\mathcal{B} \cup \nabla \mathcal{B} \cup \{+, \times\}$.

**Remark 1.** *The smallest known circuits for multiplying* $n \times n$ *matrices have size* $\Theta(n^\omega)$, *where* $\omega \approx 2.372$. *Despite tremendous research efforts only minor improvements have been obtained in recent years. There is evidence that* $\omega$ *is bounded away from 2 for certain classes of circuits [43, 44]. In contrast,* $\mathcal{S}_2$ *is computable in quadratic time* $\Theta(n^2)$ *in the basis* $B = \{\times, +, \log\}$.

The above discussion indicates that except the $\mathrm{RKE}(\mathbf{x}_1, \ldots, \mathbf{x}_n)$, i.e. order-2 Renyi entropy, whose computational complexity is quadratically growing with sample size $\Theta(n^2)$, the other members of the VENDI family $\mathrm{VENDI}_\alpha$ would have a super-quadratic complexity on the order of $\mathcal{O}(n^{2.372})$. In practice, the computation of $\mathrm{VENDI}_\alpha$ scores is performed by the eigendecomposition of the $n \times n$ kernel matrix that requires $O(n^3)$ computations for precise computation and $O(n^2 M)$ computations using a randomized projection onto an $M$-dimensional space [15, 1].

## 5 A Scalable Fourier-based Method for Computing Kernel Entropy Scores

As we showed earlier, the complexity of computing RKE and VENDI scores are at least quadratically growing with the sample size $n$. The super-linear growth of the scores' complexity with sample size $n$ can hinder their application to large-scale datasets and generative models with potentially hundreds of sample types. In such cases, a proper entropy estimation should be performed over potentially hundreds of thousands of data, where the quadratic complexity of the scores would be a significant barrier toward their accurate estimation.

Here, we consider a shift-invariant kernel matrix $k(\mathbf{x}, \mathbf{x}') = \kappa(\mathbf{x} - \mathbf{x}')$ where $\kappa(\mathbf{0}) = 1$ and propose applying the random Fourier features (RFF) framework [16] to perform an efficient approximation of the RKE and VENDI scores. To do this, we utilize the Fourier transform $\widehat{\kappa}$ that, according to Bochner's theorem, is a valid PDF, and we independently generate $\boldsymbol{\omega}_1, \ldots, \boldsymbol{\omega}_r \overset{\mathrm{iid}}{\sim} \widehat{\kappa}$. Note that in the case of the Gaussian kernel $k_{\mathrm{Gaussian}(\sigma^2)}$, the corresponding PDF will be an isotropic Gaussian $\mathcal{N}(\mathbf{0}, \frac{1}{\sigma^2} I_d)$ with zero mean and covariance matrix $\frac{1}{\sigma^2} I_d$. Then, we consider the RFF proxy feature map $\widetilde{\phi}_r : \mathbb{R}^d \to \mathbb{R}^{2r}$ as defined in (1) and define the proxy kernel covariance matrix $\widetilde{C}_{X,r} \in \mathbb{R}^{2r \times 2r}$:

$$\widetilde{C}_{X,r} = \frac{1}{n} \sum_{i=1}^{n} \widetilde{\phi}_r(\mathbf{x}_i) \, \widetilde{\phi}_r(\mathbf{x}_i)^\top \tag{2}$$

Note that the $2r \times 2r$ matrix $\widehat{C}_{X,r}$ has the same non-zero eigenvalues as the $n \times n$ RFF proxy kernel matrix $\frac{1}{n} \widetilde{K}_r$, and therefore can be utilized to approximate the eigenvalues of the original $n \times n$ kernel matrix $\frac{1}{n} K$. Therefore, we propose the *Fourier-based Kernel Entropy Approximation (FKEA)* method to approximate the RKE and $\mathrm{VENDI}_\alpha$ scores as follows:

$$\mathrm{FKEA\text{-}RKE}(\mathbf{x}_1, \ldots, \mathbf{x}_n) = \exp(H_2(\widetilde{C}_{X,r})) = \big\| \widetilde{C}_{X,r} \big\|_F^{-2}, \tag{3}$$

$$\mathrm{FKEA\text{-}VENDI}_\alpha(\mathbf{x}_1, \ldots, \mathbf{x}_n) = \exp(H_\alpha(\widehat{C}_{X,r})) = \Big( \sum_{i=1}^{2r} \widetilde{\lambda}_{r,i}^\alpha \Big)^{\frac{1}{1-\alpha}} \tag{4}$$

Note that in the above, $\widetilde{\lambda}_{r,i}^\alpha$ denotes the $i$th eigenvalue of the $2r \times 2r$ matrix $\widehat{C}_{X,r}$. We remark that the computation of both FKEA-RKE and $\mathrm{FKEA\text{-}VENDI}_\alpha$ can be done by a stochastic algorithm which computes the proxy covariance matrix (2) by summing the sample-based $2r \times 2r$ matrix terms, and then computing the resulting matrix's Frobenius norm for RKE score or all the $2r$ matrix's eigenvalues for a general $\mathrm{VENDI}_\alpha$ with $\alpha \neq 2$. Algorithm 1 presents the steps of the FKEA method where the computation needed for the proxy kernel covariance matrix is $O(n)$ and grows only linearly with sample size $n$.

Therefore, to show the FKEA method's scalability, we need to bound the required RFF size $2r$ for an accurate approximation of the original $n \times n$ kernel matrix. The following theorem proves that the needed feature size will be $\mathcal{O}\big(\frac{\log n}{\epsilon^2}\big)$ for an $\epsilon$-accurate approximations of the matrix's eigenspace.

---

**Algorithm 1** FKEA Algorithm for Computing VENDI and RKE reference-free scores

---

1: **Input:** $n$ datapoints $\mathbf{x} = \{x_1, \ldots, x_n\}$, kernel bandwidth $\sigma^2$, RFF dimension $r$
2: Draw i.i.d. samples $\boldsymbol{\omega}_1, \ldots, \boldsymbol{\omega}_r \sim \hat{\kappa}$ $\qquad\qquad\qquad$ ▷ For Gaussian Kernel $\hat{\kappa} \sim \mathcal{N}(0, \frac{1}{\sigma^2} I_d)$
3: **Initialize** the covariance matrix $\widetilde{C} \leftarrow \mathbf{0}$
4: **Compute the covariance matrix**:
5: **for** $i = 1$ to $n$ **do**
6: $\quad$ Compute the RFF feature for $x_i$:

$$\widetilde{\phi}_r(x_i) = \frac{1}{\sqrt{r}} \left[ \cos(\boldsymbol{\omega}_1^\top x_i), \quad \sin(\boldsymbol{\omega}_1^\top x_i), \quad \cdots, \quad \cos(\boldsymbol{\omega}_r^\top x_i), \quad \sin(\boldsymbol{\omega}_r^\top x_i) \right]^\top$$

7: $\quad$ Update $\widetilde{C}$:

$$\widetilde{C} \leftarrow \widetilde{C} + \frac{1}{n} \widetilde{\phi}_r(x_i)\, \widetilde{\phi}_r(x_i)^\top$$

8: **end for**
9: **Perform eigendecomposition** on the covariance matrix:

$$\{\widetilde{\lambda}_1, \ldots, \widetilde{\lambda}_{2r}\} \leftarrow \text{Eigendecomposition}(\widetilde{C})$$

10: **Compute VENDI and RKE metrics** using the eigenvalues $\widetilde{\lambda}_1, \ldots, \widetilde{\lambda}_{2r}$

---

**Theorem 2.** *Consider a shift-invariant kernel $k(\mathbf{x}, \mathbf{x}') = \kappa(\mathbf{x} - \mathbf{x}')$ where $\kappa(\mathbf{0}) = 1$. Suppose $\boldsymbol{\omega}_1, \ldots, \boldsymbol{\omega}_r \sim \widehat{\kappa}$ are independently drawn from PDF $\widehat{\kappa}$. Let $\lambda_1 \geq \ldots \geq \lambda_n$ be the sorted eigenvalues of the normalized kernel matrix $\frac{1}{n} K = \frac{1}{n} \big[ k(\mathbf{x}_i, \mathbf{x}_j) \big]_{n \times n}$. Also, consider the eigenvalues of $\widetilde{\lambda}_1 \geq \ldots \geq \widetilde{\lambda}_{2r}$ of random matrix $\widetilde{C}_{X,r}$ with their corresponding eigenvectors $\widetilde{\mathbf{v}}_1, \ldots, \widetilde{\mathbf{v}}_{2r}$. Let $\widetilde{\lambda}_j = 0$ for every $j > 2r$. Then, for every $\delta > 0$, the following holds with probability at least $1 - \delta$:*

$$\sqrt{\sum_{i=1}^{n} \big( \widetilde{\lambda}_i - \lambda_i \big)^2} \leq \sqrt{\frac{8 \log(n/2\delta)}{r}} \qquad \text{and} \qquad \sqrt{\sum_{i=1}^{n} \left\| \frac{1}{n} K \widehat{\mathbf{v}}_i - \lambda_i \widehat{\mathbf{v}}_i \right\|_2^2} \leq \sqrt{\frac{32 \log(n/2\delta)}{r}},$$

*where $\widehat{\mathbf{v}}_i := \sum_{j=1}^{r} \sin\big( \widetilde{\mathbf{v}}_{2j}^\top \mathbf{x}_i \big) \widetilde{\mathbf{v}}_{2j} + \cos\big( \widetilde{\mathbf{v}}_{2j-1}^\top \mathbf{x}_i \big) \widetilde{\mathbf{v}}_{2j-1}$ is the ith proxy eigenvector for $\frac{1}{n} K$.*

**Corollary 1.** *In the setting of Theorem 2, the following approximation guarantees hold for* RKE *and* VENDI$_\alpha$ *scores*

- *For every* VENDI$_\alpha$ *with $\alpha \geq 2$, including* RKE *for $\alpha = 2$, the following dimension-independent bound holds with probability at least $1 - \delta$:*

$$\left| \text{FKEA-VENDI}_\alpha \big( \mathbf{x}_1, \ldots, \mathbf{x}_n \big)^{\frac{1-\alpha}{\alpha}} - \text{VENDI}_\alpha \big( \mathbf{x}_1, \ldots, \mathbf{x}_n \big)^{\frac{1-\alpha}{\alpha}} \right| \leq \sqrt{\frac{8 \log(n/2\delta)}{r}}$$

- *For every* VENDI$_\alpha$ *with $1 \leq \alpha < 2$, assuming a finite dimension for the kernel feature map $\dim(\phi) = m$, the following bound holds with probability at least $1 - \delta$:*

$$\left| \text{FKEA-VENDI}_\alpha \big( \mathbf{x}_1, \ldots, \mathbf{x}_n \big)^{\frac{1-\alpha}{\alpha}} - \text{VENDI}_\alpha \big( \mathbf{x}_1, \ldots, \mathbf{x}_n \big)^{\frac{1-\alpha}{\alpha}} \right| \leq m^{\frac{1}{\alpha} - \frac{1}{2}} \sqrt{\frac{8 \log(n/2\delta)}{r}}$$

**Remark 2.** *According to the theoretical results in [45], the top-$t$ eigenvectors of kernel covariance matrix $C_X$ will correspond to the mean of the modes of a mixture distribution with $t$ well-separable modes. Theorem 2 shows for every $1 \leq i \leq 2r$, FKEA provides the proxy score function $\widetilde{u}_i : \mathbb{R}^d \to \mathbb{R}$ that assigns a likelihood score for an input $\mathbf{x}$ to belong to the ith identified mode:*

$$\widetilde{u}_i(\mathbf{x}) = \sum_{j=1}^{r} \sin\big( \widetilde{\mathbf{v}}_{2j}^\top \mathbf{x} \big) \widetilde{\mathbf{v}}_{2j,i} + \cos\big( \widetilde{\mathbf{v}}_{2j-1}^\top \mathbf{x} \big) \widetilde{\mathbf{v}}_{2j-1,i} \tag{5}$$

*Therefore, one can compute the above FKEA-based score for each of the $2r$ eigenvectors over a sample set, and use the samples with the highest scores according to every $\widetilde{u}_i$ to characterize the $i$ sample cluster captured by the FKEA method.*

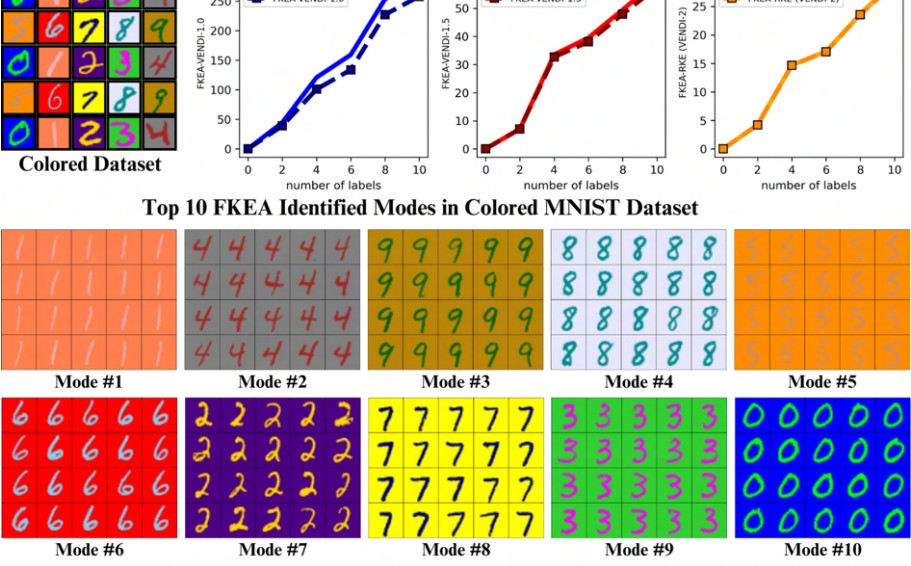

**Top 10 FKEA Identified Modes in Colored MNIST Dataset**

Figure 2: RFF-based identified clusters used in FKEA Evaluation in single-colored MNIST [46] dataset with *pixel* embedding, Fourier feature dimension $2r = 4000$ and bandwidth $\sigma = 7$. The graphs indicate increase in FKEA RKE/VENDI diversity metrics with increasing number of labels.

Table 1: Time complexity for FKEA and non-FKEA based metrics (RKE and VENDI) on ImageNet dataset with *DinoV2* embedding. Computation of VENDI and RKE on 40k+ samples are omitted due to memory overflow during metric computation.

| | | | | | | | Time (sec) | | | | | | | |
|---|---|---|---|---|---|---|---|---|---|---|---|---|---|
| | | | $2r = 8000$ | | | | | | | $2r = 16000$ | | | | |
| Metric | n=10k | n=20k | n=30k | n=40k | n=50k | n=100k | n=250k | n=10k | n=20k | n=30k | n=40k | n=50k | n=100k | n=250k |
| FKEA-RKE | 7 | 16 | 25 | 34 | 43 | 87 | 238 | 37 | 78 | 120 | 162 | 203 | 433 | 1138 |
| FKEA-VENDI | 11 | 19 | 27 | 37 | 45 | 104 | 267 | 48 | 89 | 130 | 173 | 213 | 459 | 1236 |
| RKE | 217 | 1324 | 4007 | - | - | - | - | 218 | 1330 | 4021 | - | - | - | - |
| VENDI | 286 | 1774 | 5488 | - | - | - | - | 287 | 1780 | 5502 | - | - | - | - |

# 6 Numerical Results

We evaluated the FKEA method on several image, text, and video datasets to assess its performance in quantifying diversity in different domains. In the experiments, we computed the empirical covariance matrix of $2r$-dimensional Fourier features with a Gaussian kernel with bandwidth parameter $\sigma$ tuned for each dataset, and then applied FKEA approximation for the $\text{VENDI}_1$, $\text{VENDI}_{1.5}$, and the RKE (same as $\text{VENDI}_2$) scores. An algorithm to compute these scores is presented in Algorithm 1. Experiments were conducted on RTX3090 GPUs. We interpreted the modes identified by FKEA entropy-based diversity evaluation using the eigenvectors of the proxy covariance matrix as discussed in Remark 2. For each eigenvector, we presented the training data with maximum eigenfunction values corresponding to the eigenvector.

**Time Complexity of FKEA metrics**. To highlight the computational advantages of transitioning to FKEA, Table 1 presents a comparison of the metric computations for VENDI and RKE on the ImageNet dataset, with sample sizes ranging from 10k to 250k. Our results show that VENDI and RKE become computationally intractable due to memory overflow. In contrast, the FKEA method efficiently scales up to $n = 250k$ samples, maintaining optimal computational time.

**Experimental Results on Image Data**. To investigate the FKEA method's diversity evaluation in settings where we know the ground-truth clusters and their quantity, we simulated an experiment on the colored MNIST [46] data with the images of 10 colored digits as shown in Figure 2. We evaluated the FKEA approximations of the diversity scores while including samples from $t$ digits for $t \in \{1, \ldots, 10\}$. The plots in Figure 2 indicate the increasing trend of the scores and FKEA's tight

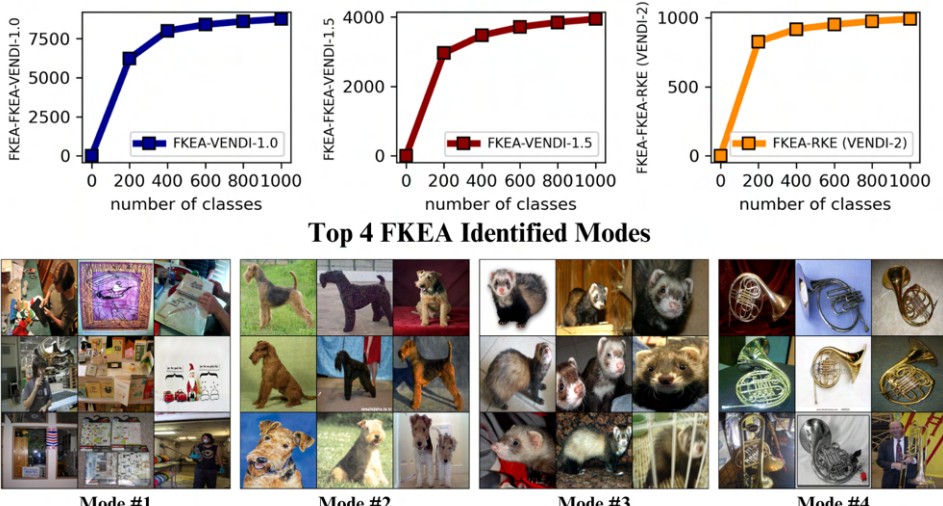

Figure 3: RFF-based identified clusters used in FKEA Evaluation in ImageNet dataset with *DinoV2* embedding, Fourier feature dimension $2r = 16k$ and Gaussian Kernel bandwidth $\sigma = 25$. The graphs indicate increase in FKEA diversity metrics with increasing number of labels per 50k samples.

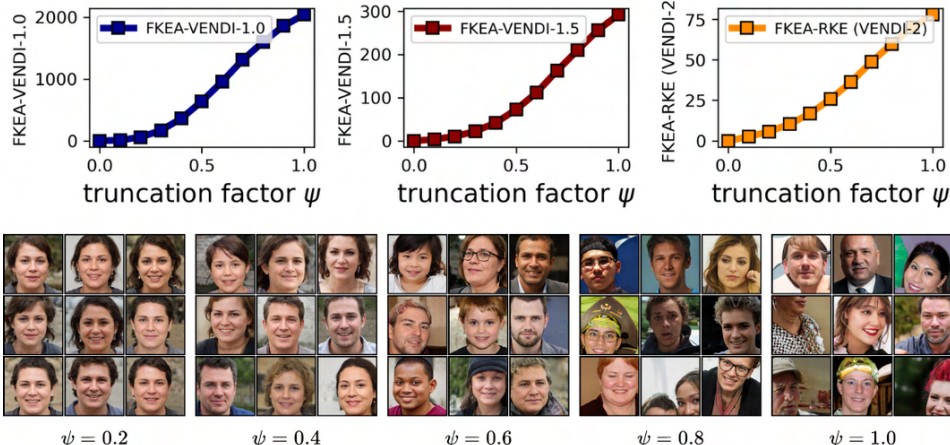

Figure 4: FKEA metrics behavior under different truncation factor $\psi$ of StyleGAN3 [47] generated FFHQ samples.

approximations of the scores. Also, we show the top-20 training samples with the highest scores according to the top-10 FKEA eigenvectors, showing the method captured the ground-truth clusters.

We conducted an experiment on the ImageNet data to monitor the scores' behavior evaluated for 50k samples from an increasing number of ImageNet labels. Figure 3 shows the increasing trend of the scores as well as the top-9 samples representing the top-4 identified clusters used for the entropy calculation. Also, Figure 4 presents the FKEA approximated entropy scores with different truncation factors in StyleGAN3 [47] on 30k generated data for each truncation factor, showing the increasing diversity scores with the truncation factor. We defer discussing the results on AFHQ [48], MS-COCO [49], F-MNIST [50] datasets to the Appendix.

Table 2: Top 5 synthetic countries dataset modes with *text-embedding-3-large* embedding, Fourier features dim $2r = 8000$ and Gaussian Kernel bandwidth $\sigma = 0.9$. The table summarises the mentions of each country in the top 100 paragraphs identified for the eigenvectors corresponding to each mode.

| Mode #1 | | Mode #2 | | Mode #3 | | Mode #4 | | Mode #5 | | Mode #6 | |
|---|---|---|---|---|---|---|---|---|---|---|---|
| Burkina Faso | 34% | Argentina | 77% | Azerbaijan | 100% | Cambodia | 94% | Belarus | 100% | Bolivia | 97% |
| Benin | 23% | Chile | 23% | | | Afghanistan | 6% | | | Ecuador | 3% |
| Chad | 22% | | | | | | | | | | |
| Burundi | 13% | | | | | | | | | | |
| Cameroon | 8% | | | | | | | | | | |

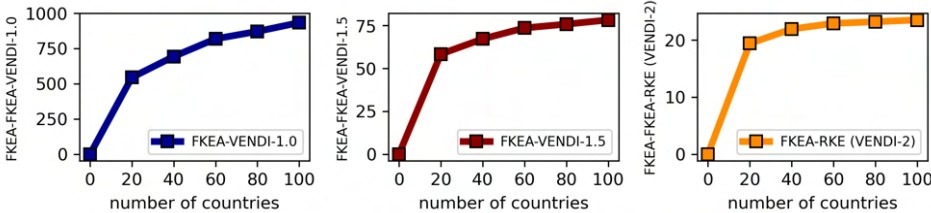

Figure 5: FKEA diversity metrics with the increasing number of countries in the synthetic dataset.

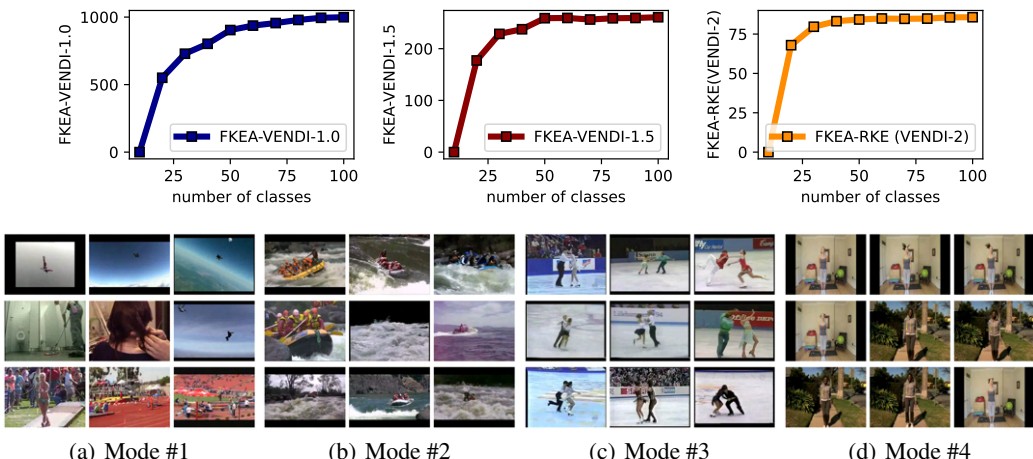

| (a) Mode #1 | (b) Mode #2 | (c) Mode #3 | (d) Mode #4 |

Figure 6: RFF-based identified clusters used in FKEA evaluation in UCF101 dataset with *I3D* embedding. The graphs indicate an increase in FKEA diversity metrics with more classes.

**Experimental Results on Text and Video Data**. To perform experiments on the text data with known clustering ground-truth, we generated 500,000 paragraphs using GPT-3.5 [51] about 100 randomly selected countries (5k samples per country). In the experiments, the text embedding used was *text-embedding-3-large* [52, 53, 51]. We evaluated the diversity scores over data subsets of size 50k with different numbers of mentioned countries. Figure 5 shows the growing trend of the diversity scores when including more countries. The figure also shows the countries mentioned in the top-6 modes provided by FKEA-based principal eigenvectors, which shows the RFF-based clustering of countries correlates with their continent and geographical location. We discuss the numerical results on non-synthetic text datasets, Wikipedia, CNN/Dailymail [54][55], CMU Movie Corpus [56], in the Appendix.

For video data experiments, we considered two standard video datasets, UCF101 [57] and Kinetics-400 [58]. Following the video evaluation literature [59, 60], we used the I3D pre-trained model [61] as embedding, which maps a video sample to a 1024-dimensional vector. As shown in Figure 6, increasing the number of video classes of test samples led to an increase in the FKEA approximated diversity metrics. Also, while the samples identified for the first identified cluster look broad, the next modes seemed more specific. We discuss the results of the Kinetics dataset in the Appendix.

## 7 Conclusion

In this work, we proposed the Fourier-based FKEA method to efficiently approximate the kernel-based entropy scores $\text{VENDI}_\alpha$ and RKE scores. The application of FKEA results in a scalable reference-free evaluation of generative models, which could be utilized in applications where no reference data is available for evaluation. A future direction to our work is to study the sample complexity of the matrix-based entropy scores and the FKEA's approximation under high-dimensional kernel feature maps, e.g. the Gaussian kernel. Also, analyzing the role of feature embedding in the method's application to text and video data would be interesting for future exploration.

## Acknowledgments

The work of Farzan Farnia is partially supported by a grant from the Research Grants Council of the Hong Kong Special Administrative Region, China, Project 14209920, and is partially supported by a CUHK Direct Research Grant with CUHK Project No. 4055164. Xuenan Cao's work is supported by a grant from the Research Grants Council of the Hong Kong Special Administrative Region, China, Project 14602223. Andrej Bogdanov's work is supported by an NSERC Discovery Grant.

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

# A Proofs

## A.1 Proof of Theorem 1

The proof of Theorem 1 combines third ingredients. The first is the relation between the circuit size of a function $C$ and of its partial derivatives $\nabla C = (\partial C/\partial x_1, \ldots, \partial C/\partial x_n)$.

**Lemma 1.** *The function $\nabla C$ has a circuit over basis $\nabla B \cup \{+, \times\}$ whose size is within a constant factor of the size of $C$.*

Lemma 1 is a feature of the backpropagation algorithm [62, 63]. This is a linear-time algorithm for constructing a circuit for $\nabla C$ given the circuit $C$ as input. In contrast, the forward propagation algorithm allows efficient computation of a single (partial) derivative even for circuits with multivalued outputs, giving the second ingredient:

**Lemma 2.** *Let $C$ be a circuit over basis $B$ and $t$ be an input to $C$. There exists a circuit that computes the derivative $\partial g/\partial t$ for every gate $g$ of $C$ over basis $\nabla B \cup \{+, \times\}$ whose size is within a constant factor of the size of $C$.*

The last ingredient is the following identity. For a scalar function $f$ over the complex numbers and matrix $X$ diagonalizable as $U\Lambda U^T$, we define $f(X)$ to be the function $Uf(\Lambda)U^T$ where $f$ is applied entry-wise to the diagonal matrix $\Lambda$.

**Lemma 3.** *For every $f$ that is analytic over an open domain $\Omega$ containing all sufficiently large complex numbers and every matrix $X$ whose spectrum is contained in $\Omega$, $\nabla \mathrm{Tr}(f(X)) = f'(X)$.*

We first illustrate the proof in the special case when $\|X\|$ is within the radius of convergence of $f$. Namely, $f(x)$ is represented by the absolutely convergent series $\sum f^{(k)}(0)x^k/k!$ for all $|x| \leq \rho$. Then $f(X) = \sum f^{(k)}(0)X^k/k!$ assuming $\|X\| \leq \rho$. By linearity (and using the fact that derivatives preserve radius of convergence) it is sufficient to show that

$$\nabla \mathrm{Tr} X^k = \frac{dX^k}{dX}, \tag{6}$$

which can be verified by explicit calculation: Both sides equal $kX^{k-1}$. This is sufficient to establish Theorem 1 for all integer $\alpha > 2$.

*Proof of Lemma 3.* The Cauchy integral formula for matrices yields the representation

$$f(X) = \frac{1}{2\pi i} \int_C f(z)(zI - X)^{-1} dz,$$

for any closed curve $C$ whose interior contains the spectrum of $X$. As $(zI - X)^{-1}$ is continuous along $C$, we can write

$$\nabla \mathrm{Tr} f(X) = \frac{1}{2\pi i} \int_C f(z) \nabla \mathrm{Tr}(zI - X)^{-1} dz. \tag{7}$$

Choosing $C$ to be a circle of radius $\rho$ larger than the spectral norm of $X$, for all $z$ of magnitude $\rho$ we have the identity

$$(zI - X)^{-1} = z^{-1}(I - z^{-1}X)^{-1} = z^{-1} \sum_{k=0}^{\infty} z^{-k} X^k$$

As the series $\sum z^{-k} \nabla \mathrm{Tr} X^k = \sum k z^{-k} X^{k-1}$ converges absolutely in spectral norm, using (6) we obtain the identity $\nabla \mathrm{Tr}(zI - X)^{-1} = d(zI - X)^{-1}/dX$, namely the lemma holds for the function $f(X) = (zI - X)^{-1}$. Plugging into (7) and exchanging the order of integration and derivation proves the lemma. $\square$

*Proof of Theorem 1.* Assume $\mathrm{Tr}\rho^\alpha$ (resp., $-\mathrm{Tr}\rho\log\rho$) has circuit size $s(d)$. By Lemma 1 and Lemma 3, $\nabla \mathrm{Tr}\rho^\alpha = \alpha\rho^{\alpha-1}$ (resp., $-\nabla \mathrm{Tr}\rho\log\rho = \log\rho - 1/\ln 2$) has circuit size $O(s(d))$. For every symmetric matrix $X$ and sufficiently small $t$, the matrix $\rho = I + tX$ is positive semi-definite. By Lemma 2 the $\mathbb{R}^{d^2}$-valued function $\partial^2\rho/\partial t^2$ has circuit size $O(s^d)$. The value of this function at

$t = 0$ is $\alpha(\alpha - 1)(\alpha - 2)X^2$ (resp., $X^2$), namely the square of the input matrix $X$ up to constant. Finally, computing the product $AB$ reduces to squaring the symmetric matrix

$$\begin{pmatrix} & A^T & B \\ A & & \\ B^T & & \end{pmatrix}. \qquad \square$$

## A.2 Proof of Theorem 2

Assuming that the shift-invariant kernel $k(\mathbf{x}, \mathbf{x}') = \kappa(\mathbf{x} - \mathbf{x}')$ is normalized (i.e. $\kappa(\mathbf{0}) = 1$), then the Fourier transform $\widehat{\kappa}$ is a valid PDF according to Bochner's theorem and also an even function because $\kappa$ takes real values. Then, we have

$$\begin{aligned} k(\mathbf{x}, \mathbf{x}') &= \kappa_\sigma(\mathbf{x} - \mathbf{x}') \\ &\overset{(a)}{=} \int \widehat{\kappa_\sigma}(\boldsymbol{\omega}) \exp(i\boldsymbol{\omega}^\top(\mathbf{x} - \mathbf{x}')) \mathrm{d}\boldsymbol{\omega} \\ &\overset{(b)}{=} \int \widehat{\kappa_\sigma}(\boldsymbol{\omega}) \cos(\boldsymbol{\omega}^\top(\mathbf{x} - \mathbf{x}')) \mathrm{d}\boldsymbol{\omega} \\ &= \mathbb{E}_{\boldsymbol{\omega} \sim \widehat{\kappa}}\left[ \cos(\boldsymbol{\omega}^\top(\mathbf{x} - \mathbf{x}')) \right] \\ &= \mathbb{E}_{\boldsymbol{\omega} \sim \widehat{\kappa}}\left[ \cos(\boldsymbol{\omega}^\top \mathbf{x}) \cos(\boldsymbol{\omega}^\top \mathbf{x}') + \sin(\boldsymbol{\omega}^\top \mathbf{x}) \sin(\boldsymbol{\omega}^\top \mathbf{x}') \right] \end{aligned}$$

Here, (a) comes from the synthesis property of the Fourier transform. (b) holds since $\widehat{\kappa_\sigma}$ is an even function, resulting in a zero imaginary term in the Fourier synthesis.

Therefore, since $|\cos(\boldsymbol{\omega}^\top \mathbf{y})| \le 1$ for all $\boldsymbol{\omega}$ and $\mathbf{y}$, one can apply Hoeffding's inequality to show for independently drawn $\boldsymbol{\omega}_1, \ldots, \boldsymbol{\omega}_r \overset{\mathrm{iid}}{\sim} \widehat{\kappa}$ the following probably correct bound holds:

$$\mathbb{P}\left( \left| \frac{1}{r} \sum_{i=1}^{r} \cos(\boldsymbol{\omega}_i^\top(\mathbf{x} - \mathbf{x}')) - \mathbb{E}_{\boldsymbol{\omega} \sim \widehat{\kappa}}\left[ \cos(\boldsymbol{\omega}^\top(\mathbf{x} - \mathbf{x}')) \right] \right| \ge \epsilon \right) \le 2\exp\left( -\frac{r\epsilon^2}{2} \right)$$

Therefore, as the identity $\cos(a - b) = \cos(a)\cos(b) + \sin(a)\sin(b)$ reveals $\frac{1}{r}\sum_{i=1}^{r} \cos(\boldsymbol{\omega}_i^\top(\mathbf{x} - \mathbf{x}')) = \widetilde{\phi}_r(\mathbf{x})^\top \widetilde{\phi}_r(\mathbf{x}')$, the above bound can be rewritten as

$$\mathbb{P}\left( \left| \widetilde{\phi}_r(\mathbf{x})^\top \widetilde{\phi}_r(\mathbf{x}') - k(\mathbf{x}, \mathbf{x}') \right| \ge \epsilon \right) \le 2\exp\left( -\frac{r\epsilon^2}{2} \right).$$

Also, $\widetilde{k}_r(\mathbf{x}, \mathbf{x}') = \widetilde{\phi}_r(\mathbf{x})^\top \widetilde{\phi}_r(\mathbf{x}')$ is by definition a normalized kernel, implying that

$$\forall \mathbf{x} \in \mathbb{R}^d : \quad \widetilde{\phi}_r(\mathbf{x})^\top \widetilde{\phi}_r(\mathbf{x}) - k(\mathbf{x}, \mathbf{x}) = 0.$$

As a result, one can apply the union bound to combine the above inequalities and show for every sample set $\mathbf{x}_1, \ldots, \mathbf{x}_n$:

$$\mathbb{P}\left( \max_{1 \le i, j \le n} \left( \widetilde{\phi}_r(\mathbf{x}_i)^\top \widetilde{\phi}_r(\mathbf{x}_j) - k_{\mathrm{Gaussian}(\sigma^2)}(\mathbf{x}_i, \mathbf{x}_j) \right)^2 \ge \epsilon^2 \right) \le 2\binom{n}{2} \exp\left( -\frac{r\epsilon^2}{2} \right).$$

Considering the normalized kernel matrix $\frac{1}{n}K = \frac{1}{n}\left[ k(\mathbf{x}_i, \mathbf{x}_j) \right]_{1 \le i, j \le n}$ and the proxy normalized kernel matrix $\frac{1}{n}\widetilde{K} = \frac{1}{n}\left[ \widetilde{\phi}_r(\mathbf{x}_i)^\top \widetilde{\phi}_r(\mathbf{x}_j) \right]_{1 \le i, j \le n}$, the above inequality implies that

$$\mathbb{P}\left( \left\| \frac{1}{n}\widetilde{K} - \frac{1}{n}K \right\|_F^2 \ge n^2 \frac{\epsilon^2}{n^2} \right) \le \binom{n}{2} \exp\left( -\frac{r\epsilon^2}{2} \right).$$

$$\implies \mathbb{P}\left( \left\| \frac{1}{n}\widetilde{K} - \frac{1}{n}K \right\|_F \ge \epsilon \right) < \frac{n^2}{2} \exp\left( -\frac{r\epsilon^2}{2} \right). \tag{8}$$

Leveraging the eigenvalue-perturbation bound in [64], we can translate the above bound to the following for the sorted eigenvalues $\lambda_1 \ge \cdots \ge \lambda_n$ of $\frac{1}{n}K$ and the sorted eigenvalues $\widetilde{\lambda}_1 \ge \cdots \ge \widetilde{\lambda}_n$ of $\frac{1}{n}\widetilde{K}$

$$\sqrt{\sum_{i=1}^{n}(\widetilde{\lambda}_i - \lambda_i)^2} \le \left\| \frac{1}{n}\widetilde{K} - \frac{1}{n}K \right\|_F$$

which shows

$$\mathbb{P}\Big(\sqrt{\sum_{i=1}^{r'}(\widetilde{\lambda}_i - \lambda_i)^2} \geq \epsilon\Big) \leq \frac{n^2}{2}\exp\Big(-\frac{r\epsilon^2}{2}\Big) \tag{9}$$

Defining $\delta = \frac{n^2}{2}\exp\Big(-\frac{r\epsilon^2}{2}\Big)$, i.e., $\epsilon = \sqrt{\frac{8\log(n/2\delta)}{r}}$, leads to

$$\mathbb{P}\Big(\sqrt{\sum_{i=1}^{n}(\widetilde{\lambda}_i - \lambda_i)^2} \leq \epsilon\Big) \geq 1 - \delta. \tag{10}$$

Noting that the normalized proxy kernel matrix $\widetilde{K}$ and the proxy kernel covariance matrix $\widetilde{C}_X$ share identical non-zero eigenvalues together with the above bound finish the proof of Theorem 2's first part.

Concerning Theorem 2's approximation guarantee for the eigenvectors, note that for each eigenvectors $\widehat{\mathbf{v}}_i$ of the proxy kernel matrix $\frac{1}{n}\widetilde{K}$, the following holds:

$$\Big\|\frac{1}{n}K\widehat{\mathbf{v}}_i - \lambda_i\widehat{\mathbf{v}}_i\Big\|_2 \leq \Big\|\frac{1}{n}K\widehat{\mathbf{v}}_i - \widetilde{\lambda}_i\widehat{\mathbf{v}}_i\Big\|_2 + \Big\|\widetilde{\lambda}_i\widehat{\mathbf{v}}_i - \lambda_i\widehat{\mathbf{v}}_i\Big\|_2$$

$$= \Big\|\Big(\frac{1}{n}K - \frac{1}{n}\widetilde{K}\Big)\widehat{\mathbf{v}}_i\Big\|_2 + |\widetilde{\lambda}_i - \lambda_i|$$

Therefore, applying Young's inequality shows that

$$\Big\|\frac{1}{n}K\widehat{\mathbf{v}}_i - \lambda_i\widehat{\mathbf{v}}_i\Big\|_2^2 \leq 2\Big\|\Big(\frac{1}{n}K - \frac{1}{n}\widetilde{K}\Big)\widehat{\mathbf{v}}_i\Big\|_2^2 + 2\big(\widetilde{\lambda}_i - \lambda_i\big)^2$$

$$= 2\mathrm{Tr}\Big(\widehat{\mathbf{v}}_i^\top\Big(\frac{1}{n}K - \frac{1}{n}\widetilde{K}\Big)^2\widehat{\mathbf{v}}_i\Big) + 2\big(\widetilde{\lambda}_i - \lambda_i\big)^2$$

$$= 2\mathrm{Tr}\Big(\widehat{\mathbf{v}}_i\widehat{\mathbf{v}}_i^\top\Big(\frac{1}{n}K - \frac{1}{n}\widetilde{K}\Big)^2\Big) + 2\big(\widetilde{\lambda}_i - \lambda_i\big)^2,$$

which implies that

$$\sum_{i=1}^{n}\Big\|\frac{1}{n}K\widehat{\mathbf{v}}_i - \lambda_i\widehat{\mathbf{v}}_i\Big\|_2^2 \leq 2\mathrm{Tr}\Big(\Big(\sum_{i=1}^{n}\widehat{\mathbf{v}}_i\widehat{\mathbf{v}}_i^\top\Big)\Big(\frac{1}{n}K - \frac{1}{n}\widetilde{K}\Big)^2\Big) + 2\sum_{i=1}^{n}\big(\widetilde{\lambda}_i - \lambda_i\big)^2$$

$$= 2\mathrm{Tr}\Big(\Big(\frac{1}{n}K - \frac{1}{n}\widetilde{K}\Big)^2\Big) + 2\sum_{i=1}^{n}\big(\widetilde{\lambda}_i - \lambda_i\big)^2$$

$$= 2\Big\|\frac{1}{n}K - \frac{1}{n}\widetilde{K}\Big\|_F^2 + 2\sum_{i=1}^{n}\big(\widetilde{\lambda}_i - \lambda_i\big)^2$$

$$\leq 4\Big\|\frac{1}{n}K - \frac{1}{n}\widetilde{K}\Big\|_F^2.$$

The above proves that

$$\mathbb{P}\Big(\sqrt{\sum_{i=1}^{n}\Big\|\frac{1}{n}K\widehat{\mathbf{v}}_i - \lambda_i\widehat{\mathbf{v}}_i\Big\|_2^2} \geq \epsilon\Big) \leq \mathbb{P}\Big(\Big\|\frac{1}{n}\widetilde{K} - \frac{1}{n}K\Big\|_F \geq \frac{\epsilon}{2}\Big)$$

$$< \frac{n^2}{2}\exp\Big(-\frac{r\epsilon^2}{8}\Big).$$

Therefore, considering the provided definition $\delta = \frac{n^2}{2}\exp\Big(-\frac{r\epsilon^2}{2}\Big)$, i.e., $2\epsilon = \sqrt{\frac{32\log(n/2\delta)}{r}}$, we will have the following which completes the proof:

$$\mathbb{P}\Big(\sqrt{\sum_{i=1}^{n}\Big\|\frac{1}{n}K\widehat{\mathbf{v}}_i - \lambda_i\widehat{\mathbf{v}}_i\Big\|_2^2} \leq 2\epsilon\Big) \geq 1 - \delta.$$

**Number of ImageNet Samples = 10k**

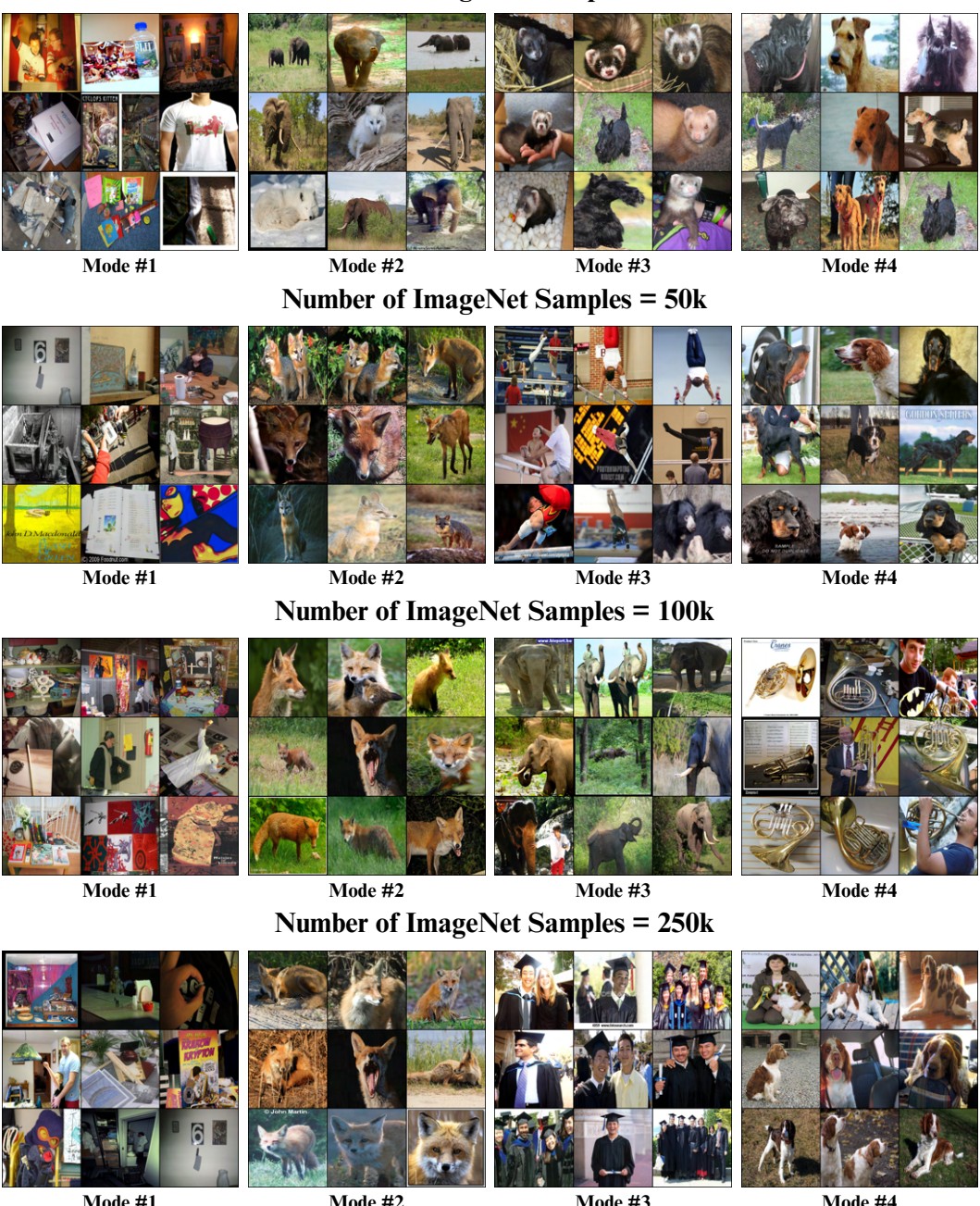

Figure 7: RFF-based identified clusters used in FKEA Evaluation in ImageNet dataset with *DinoV2* embeddings and bandwidth $\sigma = 25$ at varying number of samples $n$

**Top 8 FKEA Identified Modes in FFHQ Dataset**

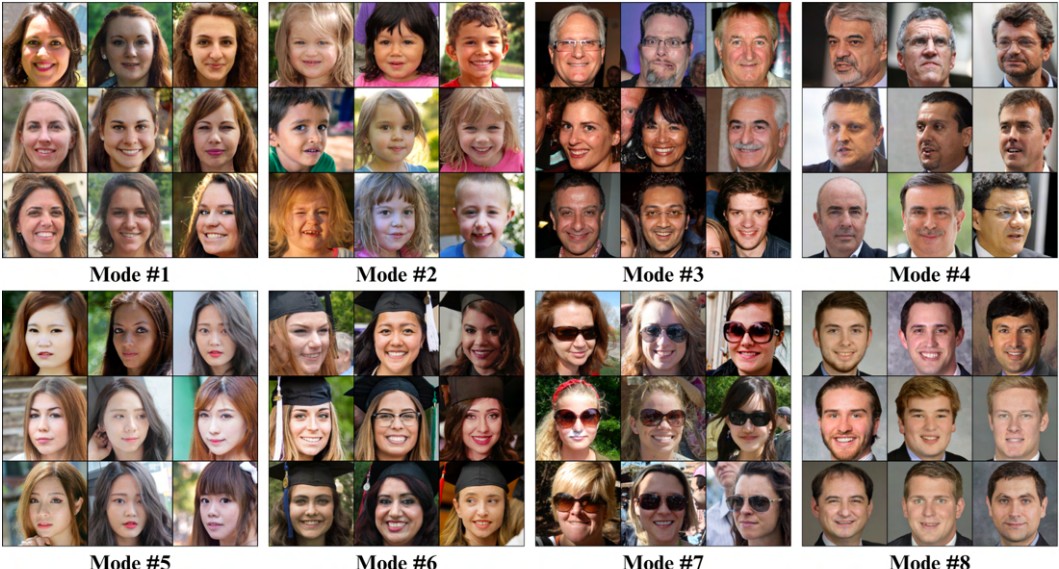

Figure 8: RFF-based identified clusters used in FKEA Evaluation in FFHQ dataset with *DinoV2* embeddings and bandwidth $\sigma = 20$

## B    Limitations

**Incompatibility with non shift-invariant kernels**. Our analysis targets a shift-invariant kernel, which does not apply to a general kernel function, such as polynomial kernels. In practice, many ML algorithms rely on simpler kernels that may not have the shift-invariant property. Due to to specifics of FKEA framework, we cannot directly extend the work to such kernels. We leave the framework's extension to other kernel functions for future studies.

**Reliance on Embeddings**. FKEA clustering and diversity assessment metrics rely on the quality of the underlying embedding space. Depending on the training and pre-training datasets, the semantic clustering properties may change. We leave in-depth study of embedding space behavior for future research.

## C    Additional Numerical Results

### C.1    Real Image Dataset Modes

This section details the results of cluster analyses conducted on various real-world datasets, including FFHQ, AFHQ, MSCOCO, and Fashion-MNIST. Each dataset's results are organized into clusters identified by RFF method in FKEA evaluation.

### C.2    The effect of number of datapoints on clustering results with FKEA

In this section, we evaluate the quality of clusters obtained from the ImageNet dataset as the number of samples $n$ varies. Specifically, we compare clustering results for 10k, 50k, 100k, and 250k samples. Figure 7 illustrates the first four modes derived from the FKEA framework.

At $n = 10k$, the clusters exhibit noise and often merge unrelated modes, as seen in Mode 2, where elephants and foxes appear within the same cluster. As $n$ increases, the clustering quality improves, becoming more coherent and meaningful. This trend is particularly evident in Modes 1 and 2, where the clusters more accurately reflect distinct semantic groups.

These findings highlight the importance of scaling VENDI and RKE scores, as computational overhead becomes a critical factor in assessing the diversity of generative models. Scaling these

## Top 8 FKEA Identified Modes in AFHQ Dataset

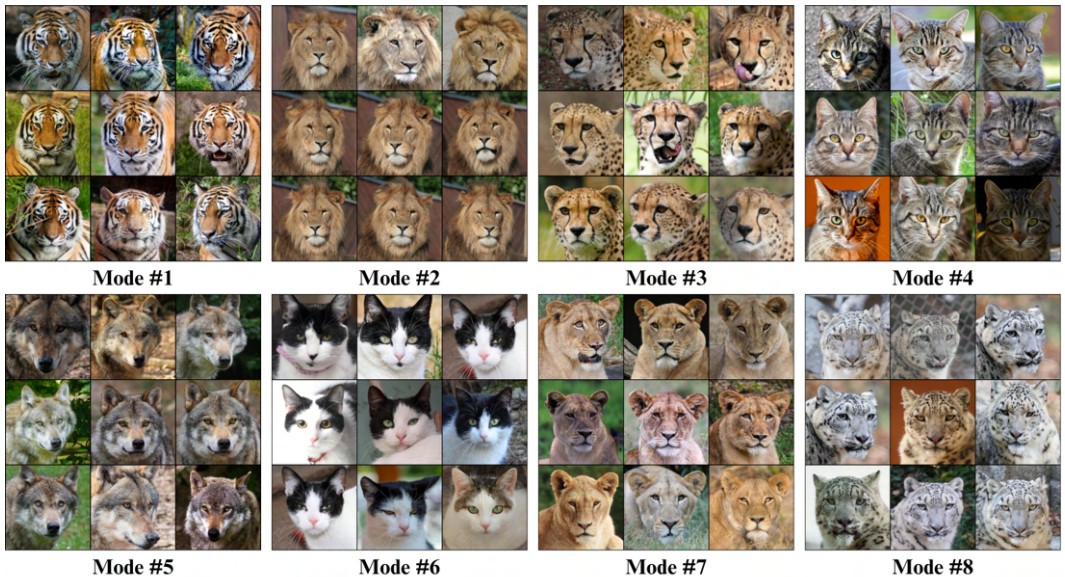

Figure 9: RFF-based identified clusters used in FKEA Evaluation in AFHQ dataset with *DinoV2* embeddings and bandwidth $\sigma = 20$

## Top 8 FKEA Identified Modes in MSCOCO Dataset

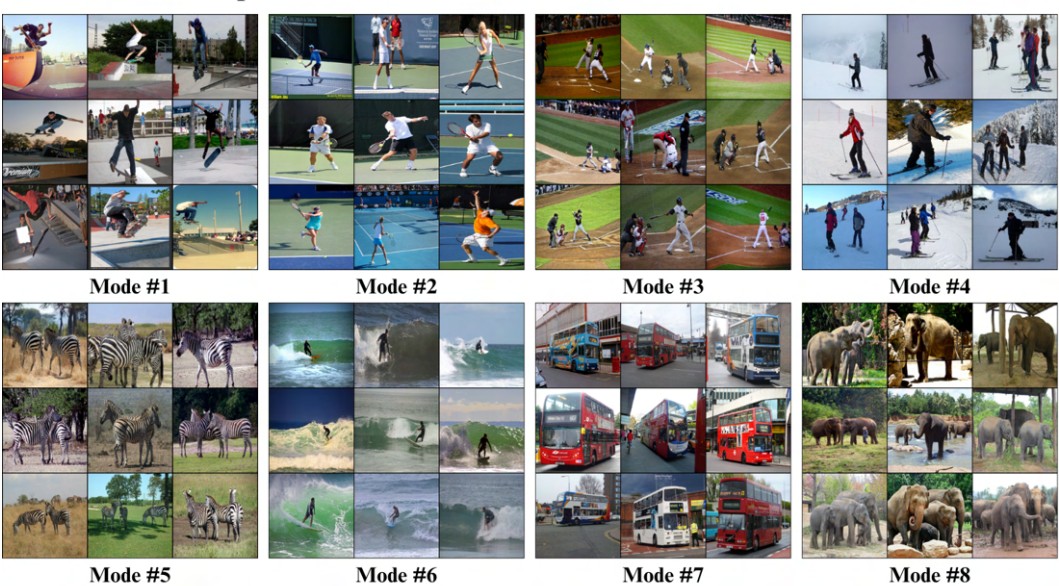

Figure 10: RFF-based identified clusters used in FKEA Evaluation in Microsoft COCO dataset with *DinoV2* embeddings and bandwidth $\sigma = 22$

**Top 8 FKEA Identified Modes in F-MNIST Dataset**

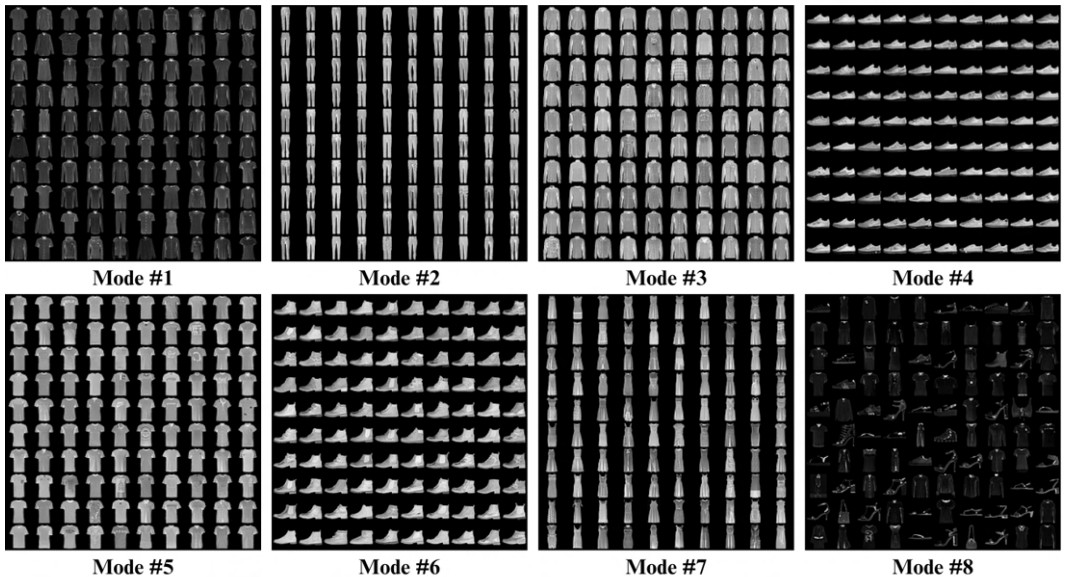

Figure 11: RFF-based identified clusters used in FKEA Evaluation in FASHION-MNIST [50] dataset with *pixel* embeddings and bandwidth $\sigma = 15$

**Top 8 FKEA Identified Modes in Color F-MNIST Dataset**

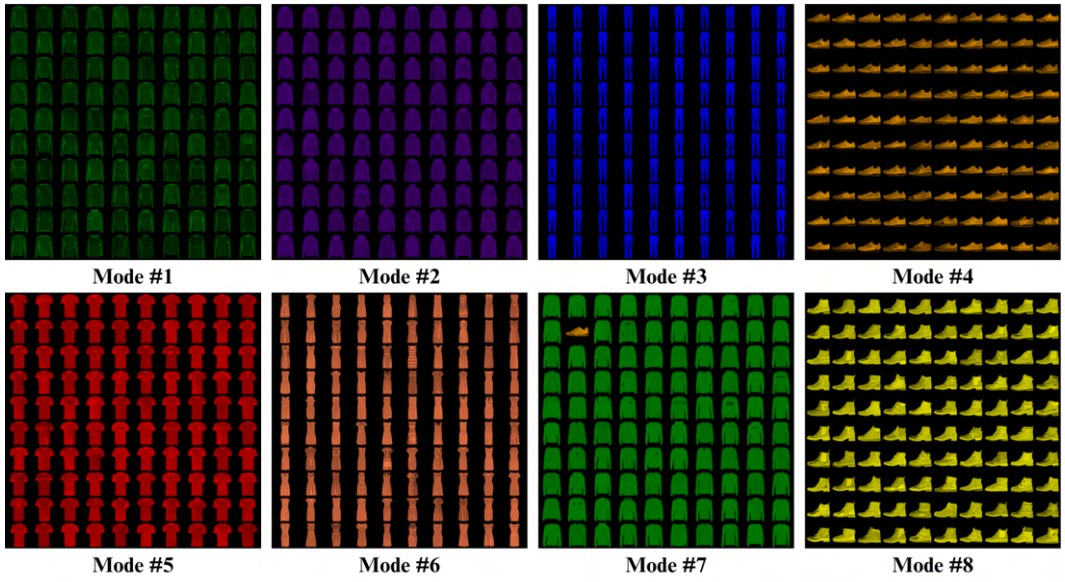

Figure 12: RFF-based identified clusters used in FKEA Evaluation in colored FASHION-MNIST [50] dataset with *pixel* embeddings and bandwidth $\sigma = 4.5$

Table 3: Evaluated scores for ImageNet generative models. The Gaussian kernel bandwidth parameter chosen for RKE, VENDI, FKEA-VENDI and FKEA-RKE is $\sigma = 25$ and Fourier features dimension $2r = 16k$. The scores were obtained by running the GitHub of [20] on pre-generated 50k samples.

| Method | IS ↑ | FID ↓ | Precision ↑ | Recall ↑ | Density ↑ | Coverage ↑ | FKEA VENDI-1 ↑ | FKEA RKE ↑ |
|---|---|---|---|---|---|---|---|---|
| Dataset (100k) | - | - | - | - | - | - | 9176.9 | 996.7 |
| ADM [65] | 542.6 | 11.12 | 0.78 | 0.79 | 0.88 | 0.89 | 8360.3 | 633.4 |
| ADMG [65] | 659.3 | 5.63 | 0.87 | 0.84 | 0.80 | 0.85 | 8524.2 | 811.5 |
| ADMG-ADMU [65] | 701.6 | 4.78 | 0.90 | 0.73 | 1.20 | 0.96 | 8577.6 | 839.8 |
| BigGAN [66] | 696.4 | 7.91 | 0.81 | 0.44 | 0.99 | 0.57 | 7120.5 | 492.4 |
| DiT-XL-2 [67] | 743.2 | 3.56 | 0.92 | 0.84 | 1.16 | 0.97 | 8626.5 | 855.8 |
| GigaGAN [68] | 678.8 | 4.29 | 0.89 | 0.74 | 0.74 | 0.70 | 8432.5 | 671.6 |
| LDM [69] | 734.4 | 4.75 | 0.93 | 0.76 | 1.04 | 0.93 | 8573.7 | 811.9 |
| Mask-GIT [70] | 717.4 | 5.66 | 0.91 | 0.72 | 1.01 | 0.82 | 8557.4 | 759.5 |
| RQ-Transformer [71] | 558.3 | 9.57 | 0.80 | 0.76 | 0.77 | 0.59 | 8078.4 | 512.1 |
| StyleGAN-XL[72] | 675.4 | 4.34 | 0.89 | 0.74 | 1.18 | 0.96 | 8171.9 | 703.5 |

metrics allows for a more efficient evaluation, especially when dealing with large datasets and high sample counts.

## C.3 Comparison between Generative Models on ImageNet dataset

In this section we report the FKEA scores for various generative models on ImageNet dataset. Table 3 evaluates the diversity scores of various ImageNet GAN models using the FKEA method applied to VENDI-1 and RKE, with potential extension to the entire VENDI family. The comparison includes baseline diversity metrics such as Inception Score [12], FID [7], Improved Precision/Recall [10], and Density/Coverage [11].

## C.4 Synthetic Image Dataset Modes

In addition to running clustering on ImageNet dataset, we also applied FKEA with varying Gaussian Kernel bandwidth parameter $\sigma$ to other datasets. The results are presented for FFHQ (Figure 8), AFHQ (Figure 9) Microsoft COCO (Figure 10) and Mono/Color versions of F-MNIST[50] (Figures 11 and 12) up to top 8 modes.

The experimental setup is similar to figure 3 with the only change is optimised bandwidth for each dataset, since datasets differ in number and typicality of the samples.

## C.5 Effect of other embeddings on FKEA clustering

Even though DinoV2 is a primary embedding in our experimental settings, we acknowldge the use of other embedding models such as SwAV[24] and CLIP[23]. The resulting clusters differ from original DinoV2 clusters and require separate bandwidth parameter finetuning. In our experiments, SwAV embedding emphasizes object placement, such as animal in grass or white backgrounds, as seen in Figure 17. CLIP on the other hand clusters by objects, such as birds/dogs/bugs, as seen in Figure 18. These results indicate that FKEA powered by other embeddings will slightly change the clustering features; however, it does not hinder the clustering performance of RFF based clustering with FKEA method.

## C.6 Effect of embeddings on score convergence

To highlight the compatibility of FKEA across diverse embedding spaces, we conducted convergence experiments on various text and image embeddings. Figure 19 presents the convergence results of the VENDI and RKE scores, comparing both FKEA and non-FKEA counterparts. Our findings show that the convergence remains consistent across different embedding spaces, demonstrating the robustness of the proposed method.

## C.7 Text Dataset Modes

To understand the applicability and effectiveness of the FKEA method beyond images, we extended our study to text datasets. We observed that clustering text data poses a more challenging task

**Top 8 FKEA Identified Modes of Generative Model LDM in FFHQ**

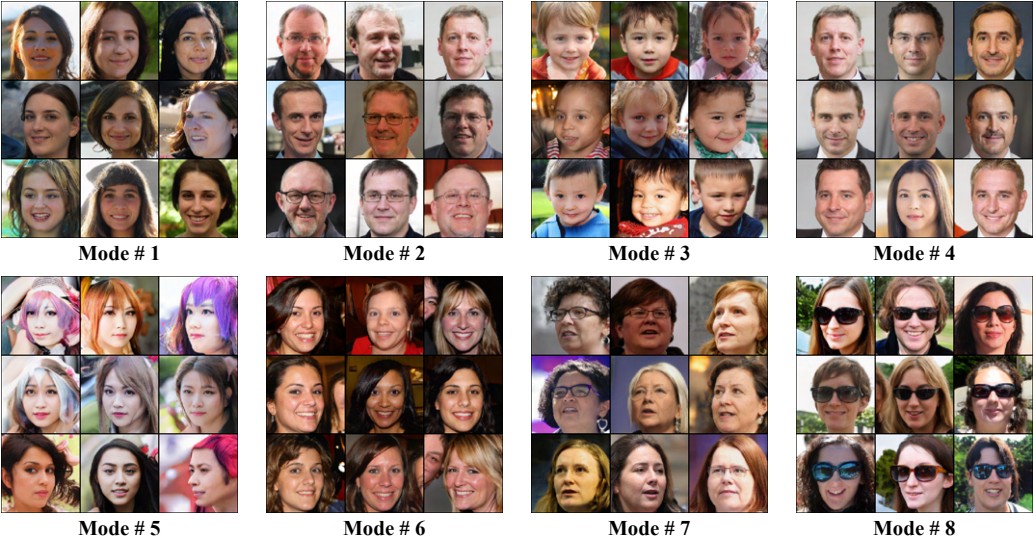

Figure 13: RFF-based identified clusters used in FKEA Evaluation of LDM [69] generative model in FFHQ with *DINOv2* embeddings and bandwidth $\sigma = 20$

**Top 8 FKEA Identified Modes of Generative Model VDVAE in FFHQ**

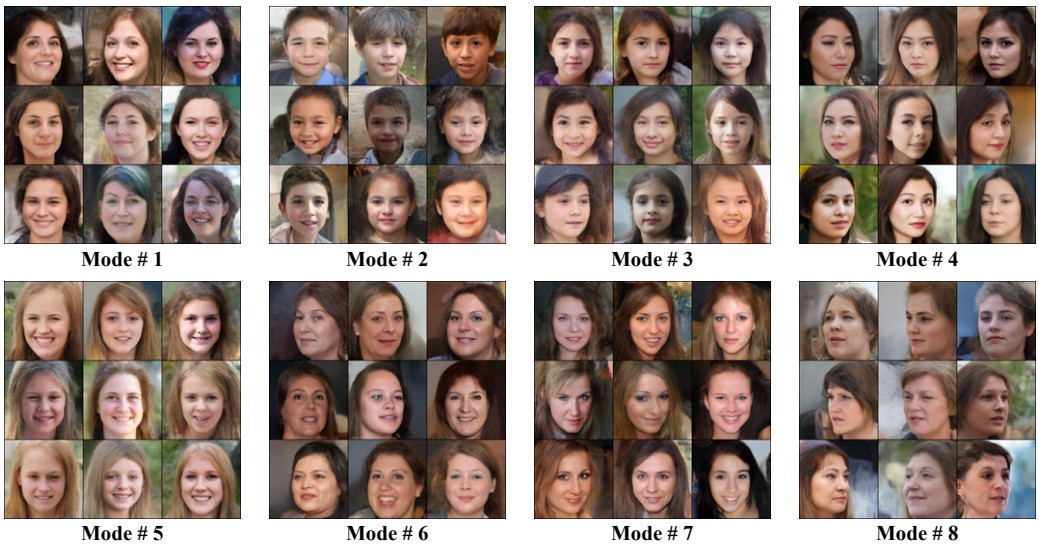

Figure 14: RFF-based identified clusters used in FKEA Evaluation of VDVAE [73] generative model in FFHQ with *DINOv2* embeddings and bandwidth $\sigma = 20$

**Top 8 FKEA Identified Modes of Generative Model InsGen in FFHQ**

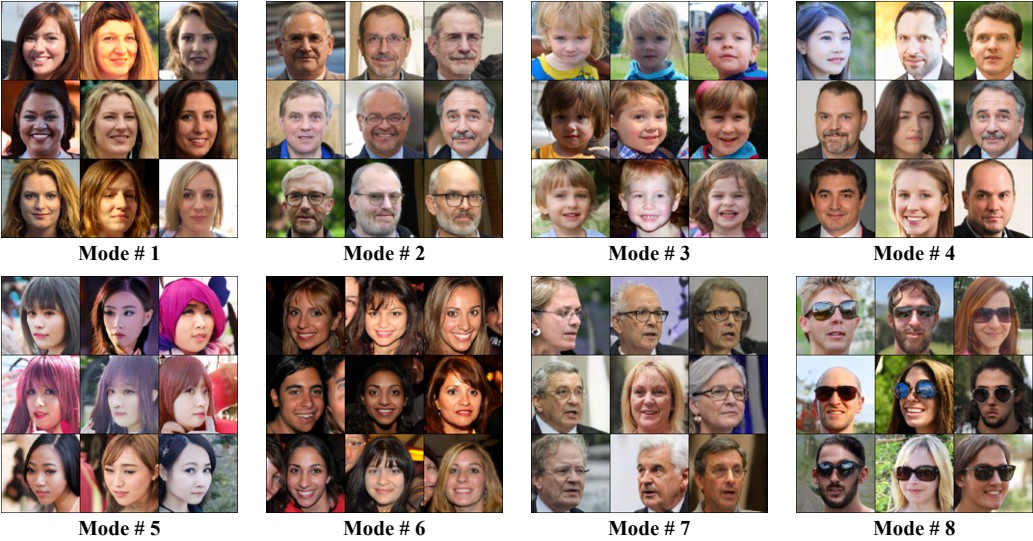

Figure 15: RFF-based identified clusters used in FKEA Evaluation of InsGen [74] generative model in FFHQ with *DINOv2* embeddings and bandwidth $\sigma = 20$

**Top 8 FKEA Identified Modes of Generative Model StyleGAN-XL in FFHQ**

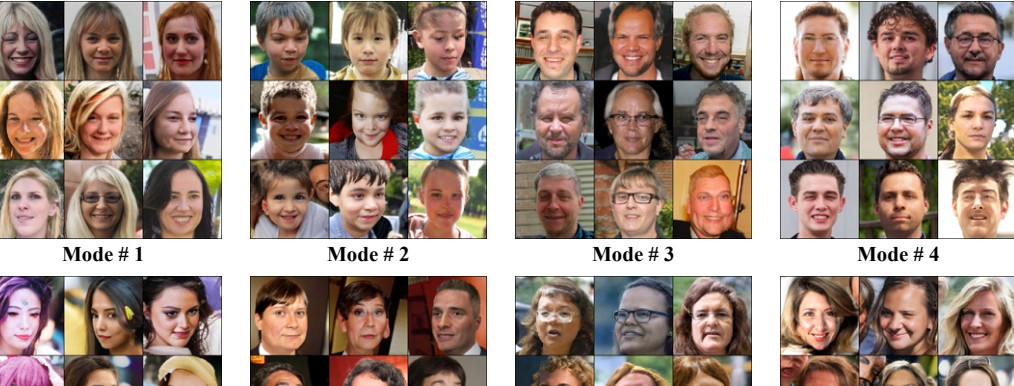

Figure 16: RFF-based identified clusters used in FKEA Evaluation of StyleGAN-XL[72] generative model in FFHQ with *DINOv2* embeddings and bandwidth $\sigma = 20$

**Top 8 FKEA Identified Modes in ImageNet Dataset with SwAV**

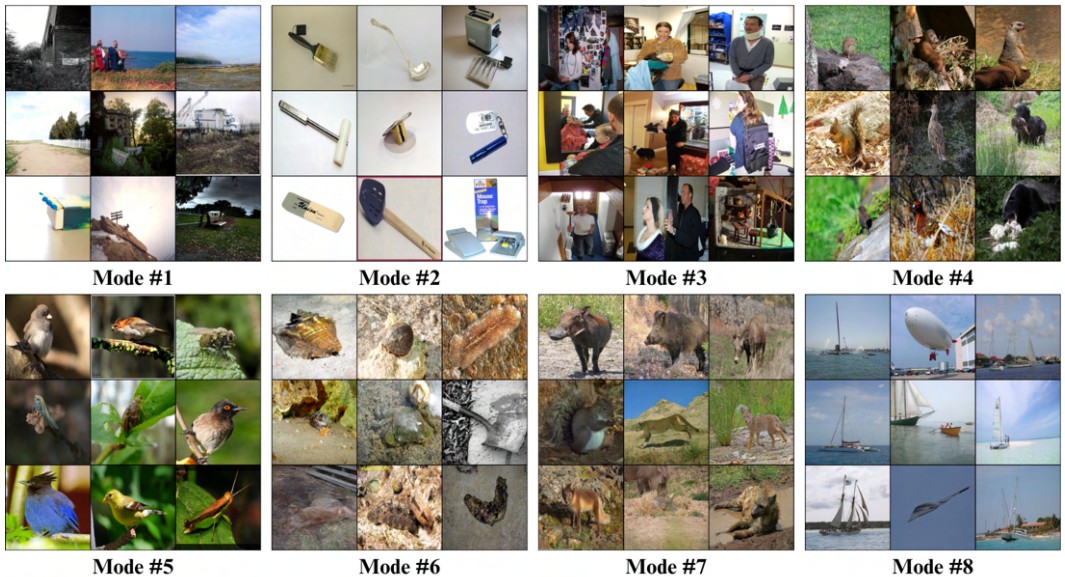

Figure 17: RFF-based identified clusters used in FKEA Evaluation of *SwAV* embedding on ImageNet with bandwidth $\sigma = 0.8$

**Top 8 FKEA Identified Modes in ImageNet Dataset with CLIP**

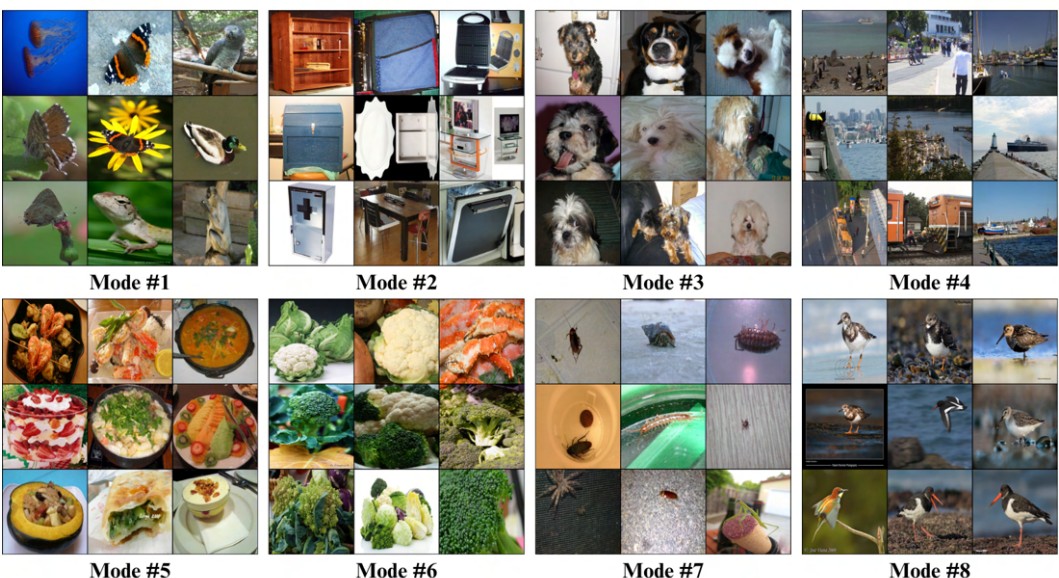

Figure 18: RFF-based identified clusters used in FKEA Evaluation of *CLIP* embedding on ImageNet with bandwidth $\sigma = 7.0$

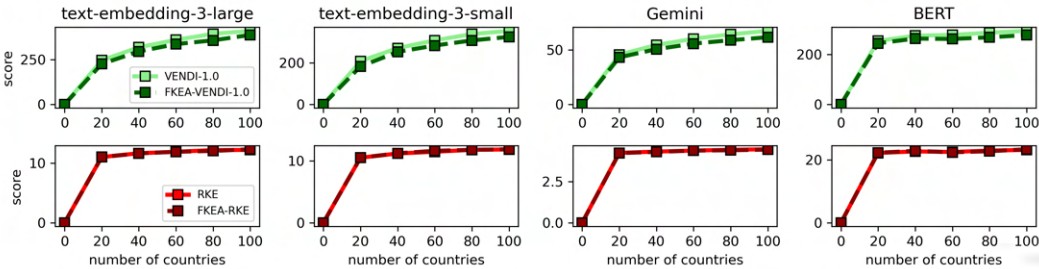

(a) Diversity convergence on synthetic countries dataset across various text embeddings

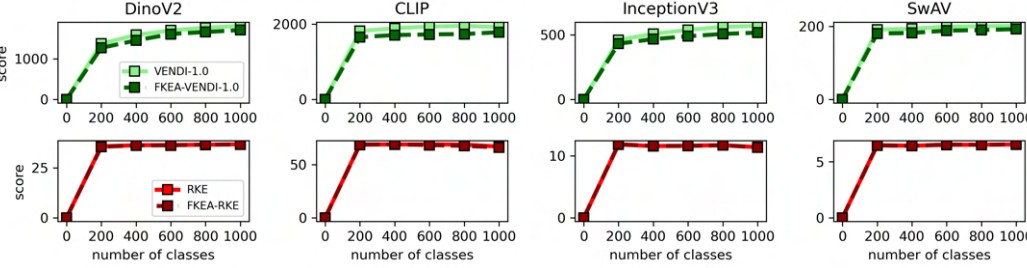

(b) Diversity convergence on ImageNet dataset across various image embeddings

Figure 19: Summary of diversity convergence with $r = 12000$ and sample size $n = 20000$.

| Mode #1 | Mode #2 | Mode #3 | Mode #4 | Mode #5 |
|---|---|---|---|---|
| Grosse Pointe | Ishkli | Gerson Garca | Girugamesh (album) | 2009 WPSL season |
| Mark Scharf | Khazora | Valentin Mogilny | Japonesque (album) | 2012 Milwaukee... |
| Alexander McKee | Sis, Azerbaijan | Gerald Lehner (referee) | Documentaly | 2020 San Antonio FC... |
| Clay Huffman | Zasun | Dmitri Nezhelev | EX-Girl | 2020 HFX Wanderers FC... |
| Ravenna, Ohio | Zaravat | Grigori Ivanov | Indie 2000 | 2020 Sporting Kansas City... |
| C. M. Eddy Jr. | Bogat | Leonidas Morakis | Triangle (Perfume album) | FC Tucson |
| Hornell, New York | Yakkakhona | Jos Luis Alonso Ber... | Waste Management (album) | 2008 K League |
| Larchmont, New York | Yava, Tajikistan | Giovanni Gasperini | Fush Yu Mang | 201112 New Zealand Football... |
| Robert Hague | Ikizyak | Mohamed Chab | Fantastipo (song) | 200809 Melbourne Victory FC... |
| General Hershy Bar | Khushikat | Louis Darmanin | Xtort | 2012 Pittsburgh Power season |
| **Keywords** | | | | |
| London | populated places | players category | music video | American football |
| American History | Maplandia.com Category | Association football | album | players Category |
| University Press | municipality | FIFA World | studio album | Football League |
| United States | village | World Cup | Records albums | League |
| World War | Osh Region | Summer Olympics | Singles Chart | League Soccer |

Table 4: Top 5 Wikipedia Dataset Modes with corresponding eigenvalues with *text-embedding-3-large* embeddings and bandwidth $\sigma = 1.0$

compared to image data. This increased difficulty arises from the ambiguity in defining clear separability factors within text, a contrast to the more visually distinguishable criteria in images. The process of evaluating text clusters is not straightforward and often varies significantly based on human judgment and perception.

To visualise the results, we use YAKE [75] algorithm to extract the keywords in each text mode and present the identified unigram and bigram keywords. We demonstrate that the results hold for text datasets and identified clusters are meaningful.

Table 4 displays the identified clusters associated with Wikipedia article titles and keywords analyzed using the FKEA method. Identified mode 1 correlates most with historical figures/events/places. Mode 2 clusters smaller villages and rural regions together. Mode 3 is exclusively about people in sports, such as athletes and referees. Mode 4 visualises various music bands and albums. Lastly, mode 5 presents the articles about sports events, such as football leagues.

Table 5 outlines the largest modes identified within a news dataset analyzed using the FKEA method, with a detailed focus on the content themes of each mode. The most dominant mode is associated with topics related to crime and police activities, indicating a frequent coverage area in the dataset.

| Mode #1 | Mode #2 | Mode #3 | Mode #4 | Mode #5 |
|---|---|---|---|---|
| police | President Obama | size | people | died |
| British police | Obama | year | severe weather | family |
| police officer | Barack Obama | weight | Death toll | plane crash |
| Police found | President | dress size | heavy rain | mother |
| family | White House | stone | Environment Agency | plane |
| found | Obama administration | Slimming World | million people | found |
| told police | Obama calls | lost | rain | people |
| court | House | lose weight | flood warnings | children |
| arrested | United States | diet | people dead | hospital |
| home | Obama plan | model | people killed | found dead |

Table 5: Top 5 CNN/Dailymail 3.0.0 [54][55] Dataset Modes with corresponding eigenvalues with *text-embedding-3-large* embeddings and bandwidth $\sigma = 0.8$.

| Mode #1 | Mode #2 | Mode #3 | Mode #4 | Mode #5 |
|---|---|---|---|---|
| The House on Tele.. | Anand | Bring Your Smile Along | Chhota Bheem... | Walk a Crooked Mile |
| Seems Like Old Times | I Love You | The Girl Most Likely | Duck Amuck | Assignment to Kill |
| Shadows and Fog | Toh Baat Pakki | Hips, Hips, Hooray! | Hare-Abian Nights | The Crime of the Century |
| Obsession | Abodh | Lady Be Good | Porky's Five and Ten | Murder at Glen Athol |
| Milk Money | Khulay Aasman... | The Courtship of Eddie's... | Sock-a-Doodle-Do | Guns |
| Very Bad Things | Kasthuri Maan... | You Live and Learn | Buccaneer Bunny | Because of the Cats |
| Blame It on the Bell... | Chhaya | Dames | Hare Lift | The House of Hate |
| The Miracle Man | Yeh Dillagi | Painting the Clouds... | Scrap Happy Daffy | The Ace of Scotland Yard |
| The Sleeping Tiger | Deva | Pin Up Girl | Hic-cup Pup | The World Gone Mad |
| The Scapegoat | Bhalobasa Bha... | Too Young to Kiss | The Goofy Gophers | Firepower |
| **Keywords** | | | | |
| mystery | hindi film | musical | animation | crime |
| noir | romance | theme songs | Tom & Jerry | murder |
| kidnap | love | city | Spike | detective |
| crime | marriage | romance | adventure | investigation |
| police | daughter | | comedy | killer |

Table 6: Top 5 CMU Movie Summary Corpus [56] Dataset Modes with corresponding eigenvalues with *text-embedding-3-large* embeddings and bandwidth $\sigma = 0.8$. The table summarises the assigned genres to each movie in the first 100 paragraphs in each mode.

Mode 2 is closely correlated with President Obama, reflecting a significant focus on political coverage. Mode 3 pertains to dieting, which suggests a presence of health and lifestyle topics. Mode 4 is linked to environmental disasters, highlighting the dataset's attention to ecological and crisis-related news. Finally, Mode 5 deals with plane crash accidents, underscoring the coverage of major transportation incidents.

Table 6 delineates the distribution of genres and production types within a dataset of movie summaries analyzed using the FKEA method. The first mode predominantly covers drama TV shows without focusing on any specific subtopic, indicating a broad categorization within this genre. From mode 2 onwards, the features become more distinct and defined. Mode 2 specifically represents Bollywood movies, with a significant emphasis on the Romance genre. Mode 3 is dedicated to clustering comedy shows. Mode 4 is exclusively associated with cartoons, evidenced by keywords such as "Tom & Jerry". Lastly, mode 5 clusters together detective and crime fiction shows.

## C.8   Video Dataset Modes

In this section, we present additional experiments on the Kinetics-400[58] video dataset. This dataset comprises 400 human action categories, each with a minimum of 400 video clips depicting the action. Similar to the video evaluation metrics, we used the I3D pre-trained model[61] which maps each video to a 1024-vector feature. Figure 20, the first mode captured broader concepts while the other models focused on specific ones. Also, the plots indicate that increasing the number of classes from 40 to 400 results in an increase in the FKEA metrics.

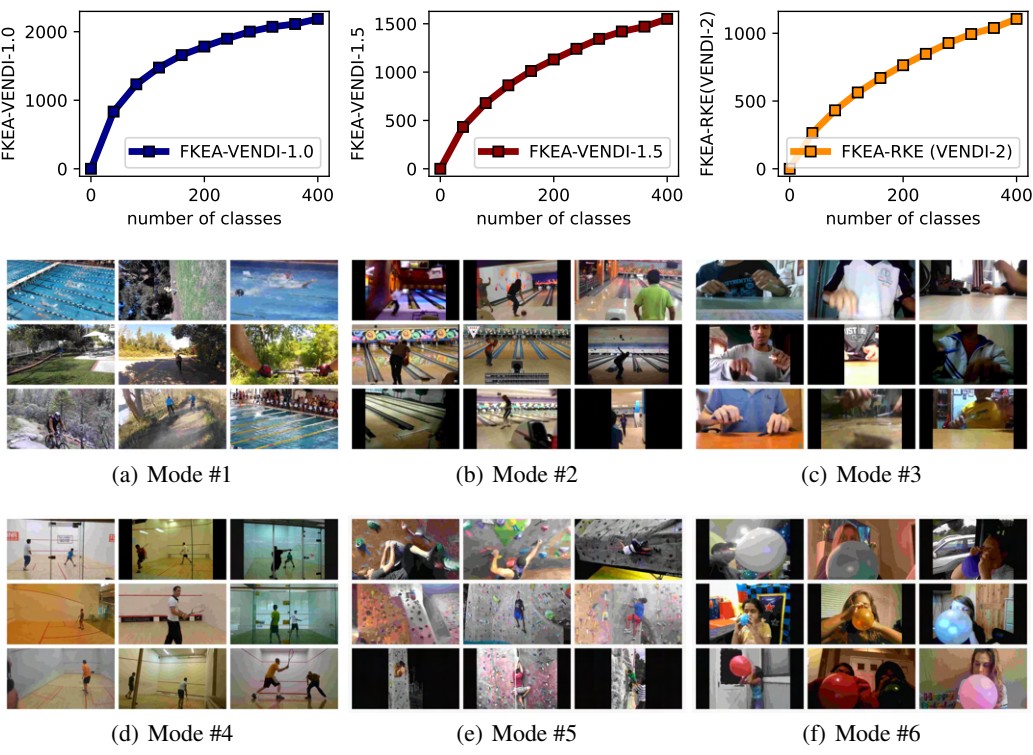

Figure 20: RFF-based identified clusters used in FKEA Evaluation in Kinetics-400 dataset with *I3D* embeddings. Plots indicate that increasing the number of classes from 40 to 400 results in an increase in the FKEA metrics.

