# OpenReview forum: "Towards a Scalable Reference-Free Evaluation of Generative Models"
_NeurIPS.cc/2024/Conference — NeurIPS 2024 poster_

### Official Review · Reviewer_ymdX · 2024-06-14

**Soundness:** 3
**Presentation:** 3
**Contribution:** 2
**Rating:** 5
**Confidence:** 4

**Summary:**

This paper introduces the Fourier-based Kernel Entropy Approximation (FKEA) metric, which efficiently evaluates the diversity of generated samples. The key contributions of this work are twofold: (1) Compared to existing diversity metrics such as VENDI and RKE, the proposed metrics (i.e, FKEA-VENDI and FKEA-RKE) are more computationally efficient, reducing complexity to $O(n)$ for a sample size $n$. (2) The proposed metric is reference-free and can be used to assess the performance of large-scale image, text, and video datasets for generative models. Extensive experimental results demonstrate the effectiveness of the proposed metric.

**Strengths:**

- The paper writing has a clear structure and the theoretical results are technically sound.
- The proposed metric exhibits improved theoretical computational efficiency (i.e., Line 13) and can be calculated without sophisticated hardware setups (Line 245).
- The method can be broadly applied to assess the performance of various generative models for images, texts, and videos.

**Weaknesses:**

- The proposed evaluation method relies on a number of parameters, e.g., $\sigma$ and $r$. The values of these parameters seem to vary significantly according to different tasks. For example, the authors adopt $2r=4,000$ and $\sigma=7$ for the experiment on the MNIST dataset, and $2r=16,000$ and $\sigma=25$ for the experiment on the ImageNet dataset. How these parameters influence the evaluation results remains unexplored. (See Questions 1 and 2)
- The paper could be improved by incorporating a toy example to illustrate the difference between reference-dependent and reference-free metrics. The discussions in Sections 1 and 2 are not concrete enough. (See Question 3)
- One of the key contributions of this paper is the theoretical improvements in the computational complexities of the FKEA-based metrics. An evaluation time comparison between FKEA-based metrics (e.g., FKEA-VENDI and FKEA-RKE) and their original metrics (e.g., VENDI and RKE) is missing. (See Question 4)

---

**Minor Point**

- There is a typo (i.e., $x \to \mathbf{x}$) in Line 157.

**Questions:**

1. The proposed metrics require stochastic approximations on expectations (i.e., $\mathbf{x}$ and $\boldsymbol{\omega}$). Could the authors report the variances of the evaluation results? Additionally, can the authors provide guidelines for choosing a sufficient number of samples?
2. Is there any underlying intuition for selecting $\sigma$ and $r$? Will an inappropriate selection of $\sigma$ and $r$ lead to an inaccurate assessment of the true generative quality?
3. Could the authors provide experimental results to demonstrate that reference-dependent metrics (e.g., Recall and Coverage) may fail to measure sample quality, while reference-free metrics may be successful?
4. Could the authors provide an evaluation time comparison between the calculation of FKEA-based metrics and their original metrics? For example, an evaluation time comparison between FKEA-VENDI and VENDI would be helpful.

**Limitations:**

The authors have allocated a section (i.e., Line 292) for the discussion of potential limitations.

---

> ### Author Rebuttal · Authors · 2024-08-07
>
> We would like to thank Reviewer ymdX for his/her time and constructive feedback and suggestions on our work. The following is our response to the reviewer’s comments and questions.
>
> **1. Variance of estimating entropy scores using FKEA**
>
> Re: To address the reviewer’s question, we measured the standard deviation of estimated FKEA entropy scores over 10 trials of running the experiment. We performed the variance estimation three times under: 1) independently drawn random Fourier features and fixed generated samples (across trials), 2) fixed (across trials) random Fourier features $\omega$ and independently drawn generated samples, 3) independently drawn random Fourier features and generated samples.
>
> The estimated entropy scores and their standard deviation are presented in Table 2 of the Rebuttal PDF. The results indicate that FKEA with $r=4000$ and $r=8000$ can yield approximations with relatively low variance.
>
>
> **2. Selection of bandwidth $\sigma$ and Fourier feature size $r$ hyperparameters**
>
> Re: We would like to clarify that the kernel bandwidth hyperparameter $\sigma$ is a specification of kernel function which is separated from the FKEA method designed to approximate the Vendi [1,3] or RKE [2] scores with the evaluator’s chosen kernel function. Therefore, in our experiments we chose the $\sigma$ parameter following References [1-3] that originally proposed the scores. Also, note that, based on the theoretical results in [2], a proper choice of hyperparameter $\sigma$ should be greater than the maximum standard deviation of the modes of a mixture distribution and be smaller than the minimum Euclidean distance between the means of different modes.
>
> On the other hand, the number of Fourier features $r$ is a hyperparameter of the FKEA method. Based on Theorem $1$, the selection $r=\frac{8(\log(n) + \log(1/2\delta))}{\epsilon^2}$ will be sufficient to guarantee an $\epsilon$-bounded approximation error for the kernel matrix’s eigenvalues with probability $1-\delta$. In our experiments, we observed that the choice of $r=8000$ for standard image datasets results in an estimation whose standard deviation is bounded by 1% of the estimated entropy.
>
>
> **3. Merits of reference-free metrics**
>
> Re: As discussed in the introduction, reference-free metrics are highly useful when the evaluator does not have access to a proper reference dataset. This can happen in scenarios where generative models produce outputs that are not represented in the standard datasets. To illustrate this, we included a toy example in Figure 1 (Rebuttal PDF) using a synthetic dataset of elephants generated by the Stable-Diffusion-XL text-to-image model. When the dataset consists of regular elephants, a reference set of “elephant” samples of ImageNet provides an accurate assessment of diversity. However, when the model is prompted to generate elephants with unrealistic colors, the internal diversity increases—something that ImageNet-based reference-dependent evaluations could not capture.
>
> In this scenario, reference-dependent metrics seemed to report either lower score (Coverage) or no significant change (Recall). In contrast, the FKEA-computed Vendi and RKE scores could identify the increased diversity, suggesting its effectiveness in evaluating generative models in contexts where traditional datasets do not fully capture the generated data's diversity. We will include this example in the revised text.
>
>
> **4. Computational costs of FKEA**
>
> Re: Table 1 in the included figures (see attached PDF to global response) summarises the improved time complexity compared to Vendi/RKE scores computed by standard PyTorch eigendecomposition method using one NVIDIA GeForce RTX 3090 GPU. The FKEA method could significantly reduce the compute and memory costs, while providing an accurate estimation of the original metric with low variance (Table 2 in Rebuttal PDF). Moreover, RKE/VENDI were too expensive to compute with moderately high sample size e.g. 40k; however, FKEA remained applicable with much higher sample sizes such as 250k.
>
> [1] Dan Friedman and Adji Bousso Dieng. The vendi score: A diversity evaluation metric for machine learning. In Transactions on Machine Learning Research, 2023.
>
> [2] Mohammad Jalali, Cheuk Ting Li, and Farzan Farnia. An information-theoretic evaluation of generative models in learning multi-modal distributions. In Advances in Neural Information Processing Systems 2023.
>
> [3] Amey Pasarkar and Adji Bousso Dieng. Cousins of the vendi score: A family of similarity-based diversity metrics for science and machine learning. In International Conference on Artificial Intelligence and Statistics 2024

---

> > ### Comment · Reviewer_ymdX · 2024-08-09
> >
> > Thank you for the response. The additional results are a nice enhancement to the paper. I have no further questions and will keep my score.

---

> > > ### Author Response · Authors · 2024-08-12
> > > **Thank you for your feedback**
> > >
> > > Dear Reviewer ymdX,
> > >
> > > We sincerely thank you for your feedback on our rebuttal. We are pleased to hear that our responses and the additional experimental results were satisfactory. As mentioned in the rebuttal, we will include these new results in the revised draft.
> > > If any further questions or comments arise during the remaining two days of the discussion period, we would be more than happy to address them.
> > >
> > > Thank you once again for your thorough review and thoughtful consideration.

---

### Official Review · Reviewer_ohKy · 2024-07-13

**Soundness:** 3
**Presentation:** 3
**Contribution:** 3
**Rating:** 5
**Confidence:** 4

**Summary:**

The work introduces a new method called Fourier-based Kernel Entropy Approximation (FKEA) to evaluate the diversity of data generated by generative models. Traditional evaluation metrics for generative models often rely on reference datasets, which may not always be available or suitable. Recently, reference-free entropy scores like VENDI and RKE have been proposed but suffer from high computational costs, especially with large-scale models.

FKEA addresses this issue by leveraging the random Fourier features framework to reduce computational complexity. It approximates the eigenspectrum of the kernel matrix to estimate entropy scores efficiently. The method utilizes proxy eigenvectors derived from FKEA to identify modes in the diversity assessment of generated samples.

**Strengths:**

1. Paper is well-written and easy to follow.
2. The contribution seems useful to the community.
3. Thorough quantitative evaluation is present on text, image and video datasets.
4. The paper mentions limitations and scope for improvement.

**Weaknesses:**

1. Evaluation seems to be limited to simple datasets. How would the method perform on complex image and video datasets?
2. The reported metrics cover the basics of establishing the advantage of this method. Any other metrics that can be used to establish clear dominance of this method over existing methods?
3. Qualitative results are hard to follow.

**Questions:**

see weaknesses section.

**Limitations:**

see weaknesses section.

---

> ### Author Rebuttal · Authors · 2024-08-07
>
> We would like to thank Reviewer ohKy for his/her time and constructive feedback on our work. The following is our response to the reviewer’s comments and questions.
>
> **1. Datasets in the numerical evaluation**
>
> In the main text, we have discussed the numerical results on the following benchmark datasets:
> - Images: MNIST, ImageNet, StyleGAN3 generated FFHQ
> - Text: Synthetic countries and landmarks dataset
> - Video: UCF101
>
> Due to the 9-page space limit, we had to defer our numerical results for the other datasets to the Appendix. We would like to refer the reviewer to sections B1, B4, B5 in the Appendix where we report our numerical results on several more large-scale datasets including FFHQ, AFHQ, MSCOCO, F-MNIST, CNN/Dailymail 3.0, Wikipedia, CMU Movie Corpus and Kinetics-400.
>
>
> **2. Comparison to other evaluation metrics in the literature**
>
> Re: We would like to clarify that our main contribution is to provide a computationally efficient method for approximating the existing reference-free Vendi and RKE evaluation scores. The Vendi and RKE metrics have been already proposed and analyzed in the literature (References [1-3]); however, they could be expensive to compute in practice. In this work, we leverage the framework of random Fourier features to reduce the computational expenses of evaluating generative models with RKE and Vendi metrics. The merits of these two metrics have been already studied in References [1-3], and as discussed in the introduction, their main application is in evaluation settings where no reference datasets are available for assessment. We refer the reviewer to Figure 1 in the Rebuttal PDF providing a toy example where the reference-based metrics did not perform satisfactorily, while the reference-free FKEA-Vendi and FKEA-RKE resulted in the proper ranking of the generative models.
>
>
>
> [1] Dan Friedman and Adji Bousso Dieng. The vendi score: A diversity evaluation metric for machine learning. In Transactions on Machine Learning Research, 2023.
>
> [2] Mohammad Jalali, Cheuk Ting Li, and Farzan Farnia. An information-theoretic evaluation of generative models in learning multi-modal distributions. In Advances in Neural Information Processing Systems 2023.
>
> [3] Amey Pasarkar and Adji Bousso Dieng. Cousins of the vendi score: A family of similarity-based diversity metrics for science and machine learning. In International Conference on Artificial Intelligence and Statistics 2024.

---

> > ### Author Response · Authors · 2024-08-12
> >
> > Dear Reviewer ohKy,
> >
> > We sincerely appreciate the time and effort you have invested in providing feedback on our work. As we approach the end of the discussion period, with only two days remaining, we wanted to ensure that all of your questions and concerns have been addressed. If there are any aspects of our submission or rebuttal that still require clarification, please let us know. We would be happy to provide any additional information or explanations.

---

> > ### Comment · Reviewer_ohKy · 2024-08-12
> >
> > Thank you for addressing my concerns. I'd stand by my initial rating.

---

### Official Review · Reviewer_NG2d · 2024-07-13

**Soundness:** 3
**Presentation:** 3
**Contribution:** 2
**Rating:** 5
**Confidence:** 4

**Summary:**

This study proposes a computationally efficient metric for evaluating the performance of recent generative models. It highlights the limitations of using reference data, which can restrict the applicability of evaluation methodologies, and instead suggests a method utilizing kernel functions without references.
Experiments demonstrate that the proposed metric shows similar trends to existing metrics across various modalities and datasets, and qualitatively reflects the characteristics of the data in the metric computation process.

**Strengths:**

* Compared to existing metrics, the proposed method allows for quick metric computation through relatively simple calculations.
* The calculation process reveals that the eigenvectors generated by the metric can semantically distinguish the generated outputs.
* It is applicable using various embedding models.

**Weaknesses:**

* Compared to existing metrics, the speed improvement in performance measurement is not experimentally specified.
* The numerical alignment with existing metrics is not provided.

**Questions:**

- Please provide the extent of the speed improvement over existing method in actual experiments.
- Provide experimental results based on the capabilities of different embedding models (e.g., compare performance measured using BERT for text with that using text-embedding-3-large).
- Measure the correlation between the existing metrics and the proposed metric in your experiments.

**Limitations:**

Yes

---

> ### Author Rebuttal · Authors · 2024-08-07
>
> We would like to thank Reviewer NG2d for his/her time and constructive feedback and suggestions on our work. The following is our response to the reviewer’s comments and questions.
>
> **1. FKEA computational costs compared to VENDI/RKE**
>
> Re: To address the reviewer’s comment, we have measured the time taken by FKEA-based and non-FKEA-based (via PyTorch eigendecomposition) computation of Vendi and RKE scores. Table 1 in the Rebuttal PDF summarizes the time complexity of FKEA evaluation compared to the baseline eigendecomposition-computed scores on one NVIDIA GeForce RTX 3090 GPU. While the baseline computation of RKE/VENDI were unaffordably expensive in terms of memory and compute power for sample sizes above 30k, FKEA-based entropy computation remained feasible at much larger sample sizes, e.g. 250k. We will add the table to the revised paper.
>
>
> **2. Effect of Embeddings on FKEA entropy evaluation**
>
> Re: Based on this comment, we performed numerical experiments and compared the Vendi/RKE diversity scores across various embeddings, including DinoV2, CLIP, SwAV, InceptionV3 for images and text-embedding-3-large/small, BERT and Gemini for text data. Figures 2a and 2b (in Rebuttal PDF) illustrate the numerical results. The results indicate that the proposed FKEA method could approximate the Vendi/RKE metrics across different embedding spaces.
>
> **3. Correlation between FKEA-approximated scores  and other diversity metrics**
>
> Re:  To address this point, we refer to Table 1 in the main text that compares FKEA-Vendi and FKEA-RKE with standard assessment metrics for diversity across several image-based generative models. The results indicate that FKEA-evaluated entropy scores correlate with existing metrics e.g. Recall and Coverage.

---

> > ### Author Response · Authors · 2024-08-12
> >
> > Dear Reviewer NG2d,
> >
> > We sincerely appreciate the time and effort you have invested in providing feedback on our work. As we approach the end of the discussion period, with only two days remaining, we wanted to ensure that all of your questions and concerns have been addressed. If there are any aspects of our submission or rebuttal that still require clarification, please let us know. We would be happy to provide any additional information or explanations.

---

### Author Rebuttal · Authors · 2024-08-07

We thank the reviewers for their constructive feedback and suggestions. We have responded to each reviewer's comments and questions under the review-box. Here we upload the 1-page PDF including the figures and plots discussed in our responses.

---

### Decision · Program_Chairs · 2024-09-25

**Decision:**

Accept (poster)

**Comment:**

The authors proposed an algorithm to evaluate the diversity of generated data in generative modelling. Most metrics used in applications rely on reference datasets and reference-free entropy scores VENDI and RKE are too computationally intensive in high dimensional settings. In this paper, the authors use the random Fourier features framework to propose an approximation of these scores reducing significantly the computational complexity.

The paper was reviewed by three experts who all gave a borderline accept score highlighting in particular that the paper is well written and that the contribution is useful for the community. The main limitation were related to the experiments to highlight the claim of the authors.

Providing efficient ways to evaluate generative models is a crucial challenge in particular for practitioners as these models are at the heart of a very intense research activity. The authors provide a very comprehensive procedure to evaluate the diversity of generated samples based on  random Fourier features supported by theoretical guarantees and extensive simulations using toy data, image and video data. The additional experiments run during the rebuttal improve the paper and confirm the reduced complexity which makes the proposed approach interesting in large scale settings where  original entropy-based diversity evaluation scores are not available.